# Correct setup of the substantia nigra requires Reelin-mediated fast, laterally-directed migration of dopaminergic neurons

Ankita Ravi Vaswani[1], Beatrice Weykopf[2†], Cathleen Hagemann[1], Hans-Ulrich Fried[3], Oliver Brüstle[2], Sandra Blaess[1]*

[1]Neurodevelopmental Genetics, Institute of Reconstructive Neurobiology, University of Bonn School of Medicine & University Hospital Bonn, Bonn, Germany; [2]Institute of Reconstructive Neurobiology, University of Bonn School of Medicine & University Hospital Bonn, Bonn, Germany; [3]Light Microscope Facility, German Center for Neurodegenerative Diseases, Bonn, Germany

**Abstract** Midbrain dopaminergic (mDA) neurons migrate to form the laterally-located substantia nigra pars compacta (SN) and medially-located ventral tegmental area (VTA), but little is known about the underlying cellular and molecular processes. Here we visualize the dynamic cell morphologies of tangentially migrating SN-mDA neurons in 3D and identify two distinct migration modes. Slow migration is the default mode in SN-mDA neurons, while fast, laterally-directed migration occurs infrequently and is strongly associated with bipolar cell morphology. Tangential migration of SN-mDA neurons is altered in absence of Reelin signaling, but it is unclear whether Reelin acts directly on migrating SN-mDA neurons and how it affects their cell morphology and migratory behavior. By specifically inactivating Reelin signaling in mDA neurons we demonstrate its direct role in SN-mDA tangential migration. Reelin promotes laterally-biased movements in mDA neurons during their slow migration mode, stabilizes leading process morphology and increases the probability of fast, laterally-directed migration.

**\*For correspondence:**
sblaess@uni-bonn.de

**Present address:** [†]Precision Neurology Program & Advanced Center for Parkinson's Disease Research, Harvard Medical School and Brigham & Women's Hospital, Boston, United States

**Competing interests:** The authors declare that no competing interests exist.

## Introduction

Dopaminergic neurons in the ventral midbrain (mDA neurons) are the major source of dopamine in the mammalian brain. Dysfunction in the dopaminergic system is associated with schizophrenia, addiction and depression, and degeneration of mDA neurons in the substantia nigra pars compacta (SN) results in the motor symptoms of Parkinson's disease (*Grace and Bunney, 1980*; *Volkow and Morales, 2015*; *Przedborski, 2017*). mDA neurons originate in the floor plate of the ventral mesencephalon, from where they migrate to cluster into the laterally-positioned SN, the medially-located ventral tegmental area (VTA) and the posterior retrorubral field. SN-mDA neurons project predominantly to the dorsal striatum and modulate voluntary movement (*Weisenhorn et al., 2016*), while VTA-mDA neurons project to various forebrain targets, including the prefrontal cortex, nucleus accumbens and basolateral amygdala, and are important for the regulation of cognitive function and reward behavior (*Morales and Margolis, 2017*). How this anatomy is setup during development remains unclear. mDA neurons differentiation starts at embryonic day (E) 10.5 in the mouse, when the first mDA neurons that express tyrosine hydroxylase (TH), the rate limiting enzyme in dopamine synthesis and a marker for differentiated mDA neurons, leave the ventricular zone of the ventral midbrain. Differentiated mDA neurons undergo a maturation process as they migrate to reach their final positions (*Blaess and Ang, 2015*). We have previously shown that both SN- and VTA-mDA neurons

undergo radial migration into the mantle layer of the developing ventral midbrain where they remain intermingled until E13.5. Between E13.5 and E15.5, mDA neurons destined for the SN migrate tangentially to more lateral positions, resulting in the segregation of mDA neurons into the laterally-located SN and the medially-situated VTA (*Bodea et al., 2014*). This particular migration pattern suggests that SN-mDA neurons have the specific molecular machinery to respond to cues in their environment that direct their lateral migration. As exemplified by migration studies in cortical brain areas, a comprehensive characterization of migratory modes and accompanying changes in cell morphology is indispensable for unraveling the molecular mechanisms by which cell-type specific migratory behavior is regulated (*Kriegstein and Noctor, 2004*). So far, a detailed understanding of mDA neuronal migratory behavior has remained elusive due to challenges in visualizing migrating mDA neurons in sufficient detail.

At the molecular level, Reelin, an extracellular matrix molecule and known regulator of neuronal migration in various brain areas, is essential for the correct lateral localization of SN-mDA neurons. Reelin binds to its receptors APOER2 and VLDLR, and induces the phosphorylation of the intracellular transducer DAB1 (*Trommsdorff et al., 1999*; *Hiesberger et al., 1999*). Phosphorylated DAB1 then mediates Reelin signaling by regulating cell adhesive properties or cytoskeletal stability (*Howell et al., 1997*; *Franco et al., 2011*; *Chai et al., 2009*). In mice homozygous for null alleles of *Reelin* (*reeler*) or *Dab1* (*scrambler* or *Dab1 null*), in *Vldlr/Apoer2* double knockout mice (*Nishikawa et al., 2003*; *Kang et al., 2010*; *Sharaf et al., 2013*), or in organotypic slices in which Reelin signaling is blocked, SN-mDA neurons do not reach their final positions in the ventrolateral midbrain and accumulate instead in the area of the lateral VTA (*Bodea et al., 2014*; *Vaswani and Blaess, 2016*). Whether Reelin affects tangential (lateral) mDA neuronal migration directly, or whether the failure of SN-mDA neurons to reach their final position in Reelin pathway mutants is due to alterations in glia fibers or neighboring neuronal populations has not been explored. Moreover, it is not understood how the loss of Reelin signaling alters dynamic migration processes of mDA neurons and which of the multiple signaling events downstream of Reelin plays a role in mDA neuronal migration.

Here, we dissect the complex dynamic morphological changes that underlie the tangential migration of SN-mDA neurons using 2-photon excitation time-lapse imaging and a semi-automated data analysis pipeline. We find that mDA neurons migrate in two modes: infrequent laterally-directed fast migration and frequent slow movement. We demonstrate that migrating mDA neurons undergo dynamic changes in cell morphology and show that fast, directed migratory spurts are strongly associated with bipolar morphology. Combining conditional gene inactivation, genetic fate mapping and time-lapse imaging, we demonstrate that Reelin affects mDA neuronal migration in a direct manner and promotes fast, laterally-directed migration of mDA neurons and stabilizes their leading process morphology.

## Results

### Reelin signaling acts directly on tangentially migrating mDA neurons

As a first step to understand the regulation of mDA tangential migration by Reelin, we investigated whether Reelin signaling is directly required by mDA neurons for their correct lateral localization. We conditionally inactivated *Dab1* in differentiated mDA neurons using a Cre-line in which Cre is knocked into the endogenous *Scl6a3 (dopamine transporter)* locus (genotype: *Scl6a3$^{Cre/+}$*, *Dab1$^{del/flox}$*; referred to as *Dab1* CKO) (*Figure 1A*; *Franco et al., 2011*; *Ekstrand et al., 2007*). To determine the onset of Cre-mediated recombination in the *Scl6a3$^{Cre/+}$* mouse line, we crossed *Scl6a3$^{Cre/+}$* mice with an enhanced yellow fluorescent protein (YFP)-expressing reporter mouse line (Rosa26$^{lox-stop-lox-EYFP}$(*Srinivas et al., 2001*). We observed widespread YFP-expression in TH-positive (TH$^+$) cells in the lateral mDA neuron domain starting at E13.5 (*Figure 1—figure supplement 1*).

*Dab1* mRNA and DAB1 protein expression was restricted to lateral mDA neurons (putative SN-mDA neurons) in the developing mDA domain at E13.5 (*Bodea et al., 2014*; *Figure 1—figure supplement 2*). To confirm the specific loss of DAB1 protein in mDA neurons of *Dab1* CKO mice, we performed immunostaining for DAB1 at E15.5. In E15.5. controls, *Dab1*/DAB1 was expressed in the forming SN and lateral VTA and in non-dopaminergic cells located laterally to the SN (*Bodea et al., 2014*; *Figure 1—figure supplement 2*). In *Dab1* CKO embryos, DAB1 was no longer expressed in

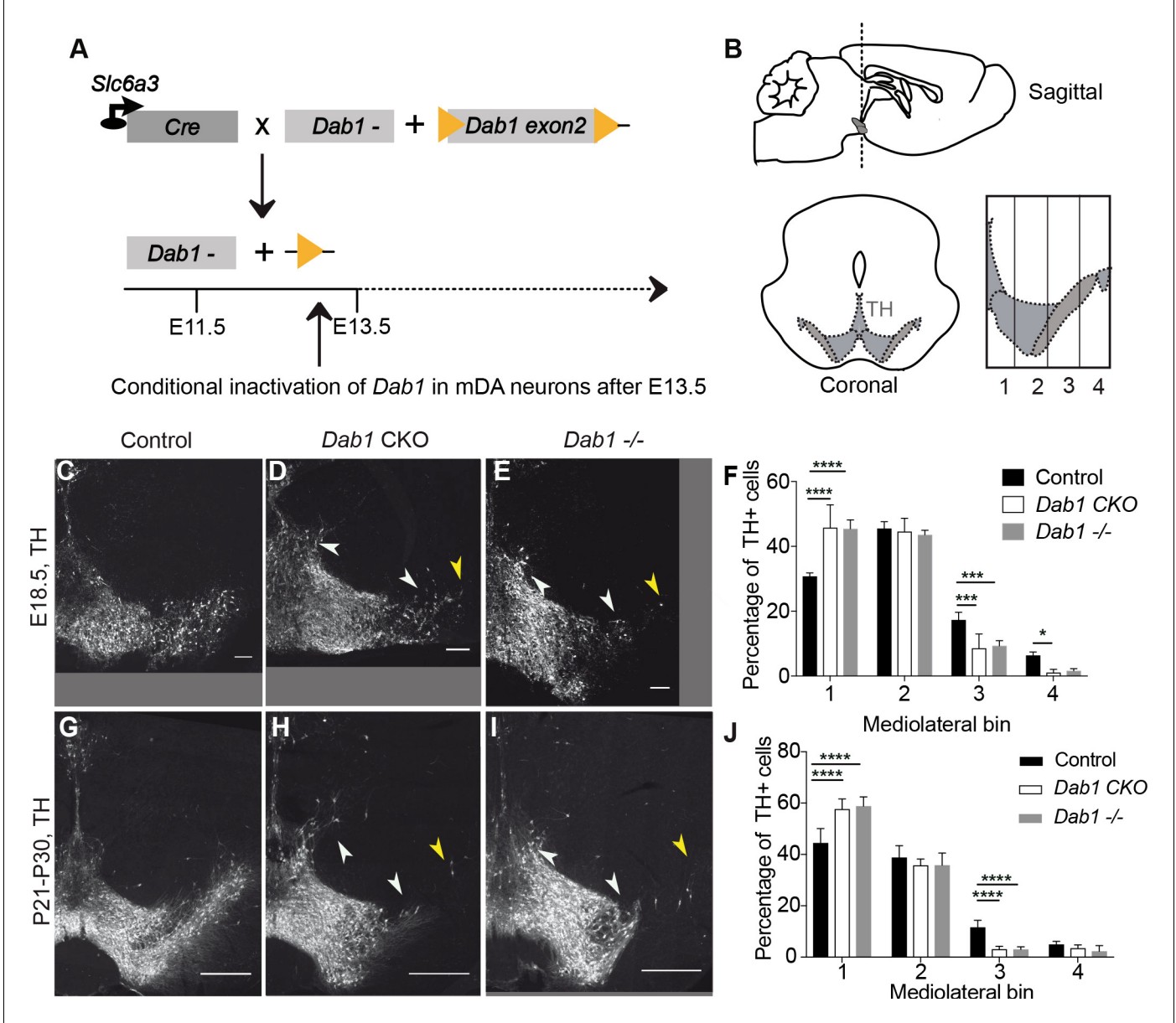

**Figure 1.** Direct role of Reelin signalling in tangential migration of mDA neurons. (**A**) Schematic showing Cre-mediated inactivation of *Dab1* in mDA neurons. (**B**) Schematic representing the anteroposterior level of coronal sections used for the analysis, and the mediolateral grid used to quantify distribution of TH$^+$ (Tyrosine Hydroxylase) neurons. (**C–J**) Immunostaining for TH and quantification of cell distribution for control, *Dab1* CKO, and *Dab1$^{-/-}$* midbrain regions at E18.5 (**C–F**) and at P21-P30 (**G–J**). White arrowheads indicate differences in the mediolateral distribution of TH$^+$ cells. Yellow arrowheads point to cells in the substantia nigra pars lateralis used as a landmark for the most lateral position in the mediolateral grids. (**F,J**) Quantification of mediolateral distribution of TH$^+$ cells for control, *Dab1* CKO and *Dab1$^{-/-}$* brains at E18.5 (**F**, n = 4 for each genotype) and at P21-P30 (**J**, n = 6 for each genotype). Data is represented as mean + s.e.m. **** indicates significant difference p<0.0001, *** indicates significant difference p<0.001, * indicates significant difference p<0.05 as assessed by two-way ANOVA with Tukey's multiple comparison correction. Scale bars: (**C–E**) 100 µm, (**G–I**) 500 µm.

The online version of this article includes the following figure supplement(s) for figure 1:

**Figure supplement 1.** *Scl6a3$^{Cre}$* mediated recombination pattern.

**Figure supplement 2.** DAB1 expression in the embryonic midbrain and specific loss of DAB1 protein in mDA neurons of *Dab1* CKO brains.

**Figure supplement 3.** Mediolateral distribution of mDA neurons at E15.5 in absence of Reelin signaling.

**Figure supplement 4.** Characterization of mDA distribution at different rostrocaudal levels in *Dab1* CKO and *Dab1$^{-/-}$* brains.

the lateral TH$^+$ domain at E15.5, while it was still present in non-dopaminergic cells (*Figure 1—figure supplement 2*). Since the inactivation of *Dab1* in *Dab1* CKO mice occurs after radial migration of SN-mDA neurons is essentially completed (*Bodea et al., 2014*), any defects observed in SN formation in this mouse model can be attributed to misregulation of mDA tangential migration. DAB1 has recently been reported to act as an effector downstream of Netrin and its receptor deleted in colorectal cancer (DCC) (*Zhang et al., 2018*), but given that *Dab1* null, *Apoer2/Vldlr* double mutant and *reeler* mice have very similar phenotypes in mDA neuron development (*Nishikawa et al., 2003*; *Kang et al., 2010*; *Sharaf et al., 2013*), we assume that the phenotype caused by the specific inactivation of *Dab1* in mDA neurons will primarily reflect its function downstream of Reelin signaling. Thus, the *Dab1* CKO model allows us to dissect out the direct role of Reelin signaling in the tangential migration of mDA neurons.

SN-mDA neurons fail to migrate to their correct lateral position in *reeler*, *Dab1* null or *Apoer2/Vldlr* double knock-out mutants (*Nishikawa et al., 2003*; *Kang et al., 2010*; *Bodea et al., 2014*). To examine whether this phenotype is recapitulated in *Dab1* CKO mice, we compared the mediolateral distribution of TH$^+$ mDA neurons in coronal midbrain sections of control, *Dab1* CKO and *Dab1$^{-/-}$* (genotype: *Dab1$^{del/del}$*) mice at postnatal day (P)21-P30 and embryonic time points (E15.5 and E18.5) (*Figure 1*, *Figure 1—figure supplement 3*, *Figure 1—figure supplement 4*). Four rostrocaudal levels were analyzed in postnatal mice showing a disorganization in the arrangement of mDA neurons, in particular in the SN, in both *Dab1* CKO and *Dab1$^{-/-}$* mice (*Figure 1G–I*, *Figure 1—figure supplement 4A–I*). In addition, a few mDA neurons were aberrantly located dorsolateral to the VTA in *Dab1* CKO and *Dab1$^{-/-}$* brains (*Figure 1G-I*, *Figure 1—figure supplement 4G–I*). The SN phenotype was most severe at the most rostral level and at an intermediate rostrocaudal level of the mDA-containing region (*Figure 1G–I*, *Figure 1—figure supplement 4A–C*). At the intermediate rostrocaudal level, mDA neurons failed to reach lateral positions in the SN and settled in more medial locations in both *Dab1* CKO and *Dab1$^{-/-}$* mice, a phenotype that was already evident at E15.5 and E18.5 (*Bodea et al., 2014*; *Figure 1C–J*, *Figure 1—figure supplement 3*). Analysis of the distribution of mDA neurons in the anatomically defined SN and VTA area showed that there was a significant shift of mDA neurons from the SN to the VTA at the intermediate rostrocaudal level. The loss of cells located in lateral SN region (Control 39.6% SN, 60.4% VTA; *Dab1* CKO: 23.9% SN, 76.1% VTA; *Dab1$^{-/-}$*: 25.6% SB, 74% VTA) was roughly equal to the gain of cells in the VTA region (*Figure 1—figure supplement 4K–M*). As the shift in the mediolateral distribution of mDA neurons observed in *Dab1* CKO and *Dab1$^{-/-}$* brains was similar, we conclude that Reelin acts directly on SN-mDA neurons to regulate their lateral migration.

We then asked whether such a direct function of Reelin is consistent with the localization of Reelin protein. During the time window of SN-mDA tangential migration (before E15.5), *Reelin* mRNA is expressed in the red nucleus, which is located dorsomedial to SN-mDA neurons. Whether Reelin protein is localized close to migrating SN-mDA neurons during this period has not been investigated (*Nishikawa et al., 2003*; *Sharaf et al., 2015*; *Bodea et al., 2014*). Immunostaining for Reelin at E13.5 and E14.5 confirmed strong expression of the protein in the region of the red nucleus (*Figure 2B,C,E,F*). At E13.5 and E14.5, Reelin protein, but not *Reelin* mRNA, was also observed ventral and lateral to the red nucleus, including the area where the most lateral mDA neurons are localized at these stages (*Figure 2A–G*). Thus, the localization of Reelin protein at E13.5-E14.5 is consistent with a direct role of Reelin signaling in SN-mDA neuronal migration.

## Reelin signaling contributes to the segregation of SN- and VTA-mDA neurons into separate clusters

Given that SN-mDA neurons fail to form the lateral SN in the absence of Reelin signaling, we asked whether Reelin signaling is important for the segregation of SN- and VTA-mDA neurons into separate clusters. We have previously shown that mDA neurons positive for the potassium channel GIRK2 (G-protein-regulated inward-rectifier potassium channel 2; expressed in mDA neurons in the SN and lateral VTA) are shifted medially in *Dab1$^{-/-}$* mice, while mDA neurons positive for Calbindin (expressed in VTA-mDA neurons and in a dorsal subset of SN-mDA neurons) are correctly localized (*Bodea et al., 2014*; *Björklund and Dunnett, 2007*). Comparison of the mediolateral position of TH$^+$, Calbindin$^+$ and TH$^+$, GIRK2$^+$ cells in control and *Dab1* CKO brains at P30 showed that there was no significant difference in the distribution of TH$^+$, Calbindin$^+$ mDA neurons between *Dab1* CKO mice and controls (data not shown). In contrast, the TH$^+$, GIRK2$^+$ mDA subpopulation showed

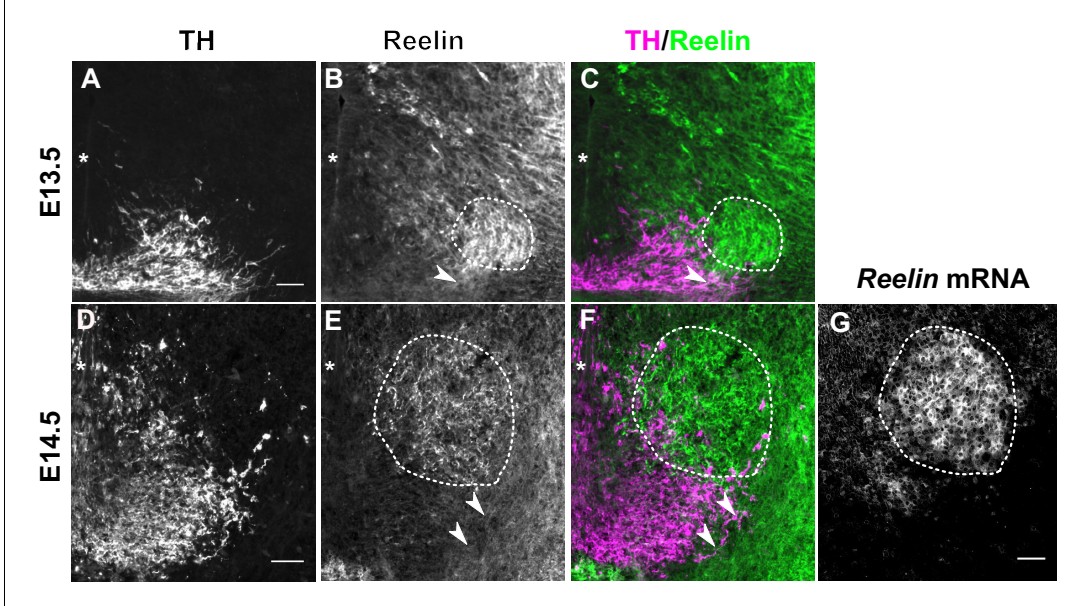

**Figure 2.** Reelin protein in the ventral midbrain at E13.5 and E14.5. (A–C) Double immunolabeling for TH and Reelin shows Reelin protein in the region of the red nucleus (RN, dashed outline) and in the lateral TH⁺ mDA domain (arrowhead) at E13.5. (D–G) Double immunolabeling for TH and Reelin (D–F) and RNA in situ hybridization for *Reelin* mRNA (G) at E14.5. *Reelin* mRNA and Reelin protein are strongly expressed in the RN. Reelin protein is also localized ventral and lateral to the RN, overlapping with the lateral mDA domain (arrowheads). Note that a brightfield image was acquired for the in situ hybridization signal of *Reelin*. The brightfield image was inverted to obtain the image shown in (G). Asterisks indicate ventral midline. Scale bar: 50 μm.

a significant shift to a more medial position in the *Dab1* CKO mice (*Figure 3A–C*). These results further confirmed that the *Dab1* CKO phenotype recapitulates the phenotype observed in *Dab1⁻/⁻* mice.

To investigate the distribution of medially shifted SN-mDA neurons within the VTA we analyzed the expression of the transcription factor SOX6 (sex determining region Y-box6), and the Lim domain protein LMO3 (LIM domain only protein 3) as markers for SN-mDA neurons and the expression of the transcription factor OTX2 (Orthodenticle homeobox 2) in VTA-mDA neurons (*Di Salvio et al., 2010*; *Panman et al., 2014*; *Poulin et al., 2014*; *La Manno et al., 2016*; *Bifsha et al., 2017*; *Poulin et al., 2018*). In E18.5 control brains, TH⁺, OTX2⁺ cells and TH⁺, SOX6⁺ cells were clearly separated at the boundary between SN and lateral VTA and TH⁺, *Lmo3*⁺ mDA neurons were located in the SN (*Figure 3D,F,H*). In *Dab1* CKO mice, TH⁺, SOX6⁺ and TH⁺, *Lmo3*⁺ mDA neurons were more medially located than in controls and were partially intermingled with TH⁺, OTX2⁺ mDA neurons at the border between SN and lateral VTA (*Figure 3E,G,I*). Hence, the inactivation of Reelin signaling in mDA neurons results in an ectopic medial location of SN-mDA neurons and a partial mixing of the two populations at what would constitute the SN-lateral VTA border in control brains.

Finally, to understand whether the altered mediolateral distribution of mDA neurons in *Dab1* CKO and *Dab1⁻/⁻* mice results in defects in axonal outgrowth and innervation of forebrain targets, we analyzed axonal projections in whole-mount control and *Dab1⁻/⁻* brains at E13.5. In both control and *Dab1⁻/⁻* embryonic brains, TH⁺ fibers extended from the midbrain to the forebrain (*Figure 3—figure supplement 1A-D*). At P30, the density of TH⁺ fibers was similar in the striatum of control and *Dab1⁻/⁻* brains (*Figure 3—figure supplement 1E-G*, three rostrocaudal levels). Hence, loss of Reelin signaling and the consequent changes in the anatomical organization of mDA neuronal cell bodies do not seem to affect general axon pathfinding processes in mDA neurons.

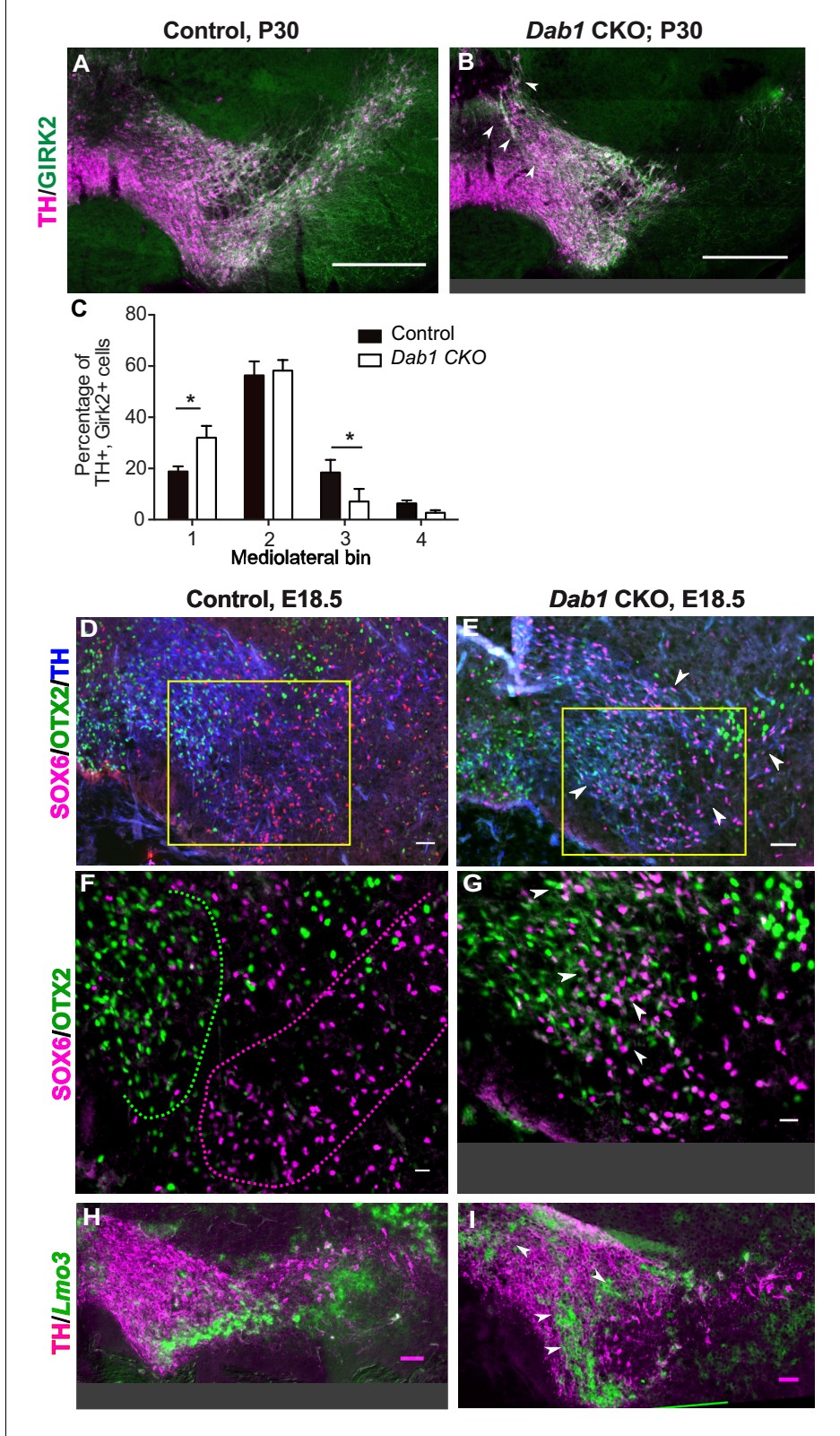

**Figure 3.** SN-mDA neurons do not completely segregate from VTA-mDA neurons at the SN-lateral VTA border in the absence of Reelin signaling. (A–C) Immunostaining for TH and GIRK2 in P30 *Dab1* CKO mice. Arrowheads: medial accumulation of TH[+], GIRK2[+] cells in *Dab1* CKO (B) compared to control littermates (A). TH[+], GIRK2[+] cells are shifted medially in *Dab1* CKO brains compared to controls (C). * indicates p<0.05 as assessed by Student's

*Figure 3 continued*

t-test corrected for multiple comparisons (Holm-Sidak method) for n = 3 brains/genotype. (D,E) Triple immunostaining for SOX6 (magenta), OTX2 (green) and TH (blue) on E18.5 control and *Dab1* CKO brains. Arrowheads indicate altered distribution of TH+, SOX6+ cells in *Dab1* CKO mice. Boxed areas indicate regions presented in F and G. (F,G) Higher zoom of TH+ lateral VTA region shown in D and E. In controls, SOX6+ cells (dashed magenta line) and OTX2+ cells (dashed green line) are largely localized to separate regions (F). In *Dab1* CKO mice, SOX6+ cells accumulate medially and are partially intermingled with OTX2+ cells at the SN-lateral VTA border (arrowheads) (G). (H,I) Immunostaining for TH and RNA in situ hybridyzation for *Lmo3* shows ectopic localization of TH+, *Lmo3*+ cells in *Dab1* CKO mice (arrowheads in I). Note that the signal for *Lmo3* expression was inverted and then false-colored in green. Cells, in which *Lmo3* was detected, show weak TH immunostaining, since the strong RNA in situ hybridization signal interferes with antibody binding. Scale bars: (A,B) 200 μm, (D,E,H,I) 50 μm, (F,G) 25 μm.

The online version of this article includes the following figure supplement(s) for figure 3:

**Figure supplement 1.** Characterization of mDA projections in the absence of Reelin signaling.

## Time-lapse imaging of tangentially migrating mDA neurons reveals diverse migratory behaviors across a population of neurons, and in individual neurons across time

Having established the direct requirement of Reelin signaling in the tangential migration of SN-mDA neurons, we visualized their migration in the presence and absence of Reelin, thereby dissecting out the precise migratory behaviors regulated by Reelin signaling. To monitor mDA migration during development, sparse labeling of SN-mDA neurons is necessary to enable tracking and morphology analysis of their migration. We used an established genetic inducible fate mapping system to mosaically label SN-mDA progenitors and their descendants (*Figure 4A*; *Blaess et al., 2011*; *Bodea et al., 2014*). With this system, SN-mDA neurons are preferentially labeled and more than two-thirds of YFP-labeled neurons are TH+ in the imaged regions at E13.5, and almost 90% are TH+ at E14.5 (*Blaess et al., 2011*; *Bodea et al., 2014*). Henceforth, we refer to these YFP-labeled neurons as SN-mDA neurons.

Ex vivo horizontal organotypic slice cultures of the ventral brain from E13.5 embryos with mosaically labelled SN-mDA neurons were prepared for time-lapse imaging (*Figure 4B*; *Bodea and Blaess, 2012*; *Bodea et al., 2014*). As horizontal slices were used for the imaging, prospective mDA neurons along the rostrocaudal axis of the forming mDA region were present in these slices. 2-photon excitation time-lapse microscopy allows 3D visualization of dynamic changes in cell morphologies of migrating SN-mDA neurons. As the migratory modes and associated changes in morphology of tangentially-migrating mDA neurons are unknown, we first defined migratory behavior in SN-mDA neurons using a number of parameters in slices of control mice and subsequently compared them with those of SN-mDA neurons in *Dab1*-/- slices.

To characterize the whole range of migratory behaviors within the time window of imaging, we acquired 3D volume images of slices every 10 min and tracked soma positions of a large number of neurons (806 neurons from three control slices, 844 neurons from 3 *Dab1*-/- slices). We then calculated speed and trajectory for each neuron's soma, at every time-point of imaging, based on location differences in consecutive volume images (*Figure 4C–F*, *Video 1*). Plotting average speed distributions of cells from each slice, showed that the behavior of cells in different control slices and in different *Dab1*-/- slices was comparable (*Figure 4—figure supplement 1A,B*). However, individual cells' soma speeds varied considerably over time, and the maximum observed soma speed (henceforth max-speed) of a cell could be several times higher than its average speed (*Figure 4F*). Furthermore, ranking all control and all *Dab1*-/- cells by their max-speeds revealed great diversity as the max-speeds varied across cells in a smooth distribution from 183 μm/hr to 0 μm/hr for controls and from 134 μm/hr to 0 μm/hr for *Dab1*-/- cells (*Figure 4—figure supplement 1C,D*).

## Two modes of tangential migration in SN-mDA neurons: frequent, slow movements and infrequent, fast movements that are promoted by Reelin signaling

The role of Reelin signaling has been studied extensively in the cortex and hippocampus. However, only few studies have examined Reelin function in regulating the speed of migrating neurons. These

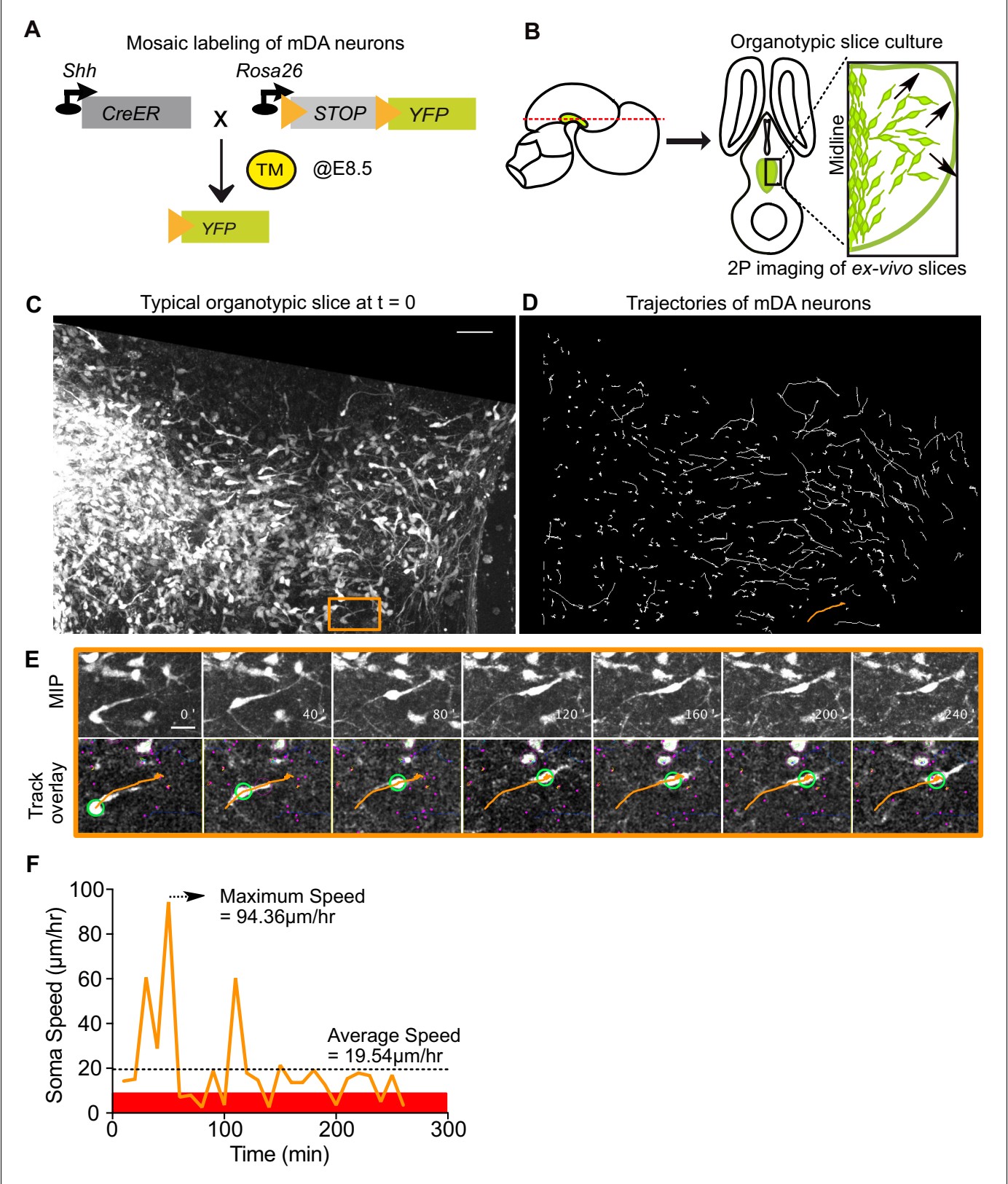

**Figure 4.** Visualizing mDA tangential migration with 2-photon excitation microscopy. (A) Schematic of the inducible genetic fate mapping system used to mosaically label mDA neurons by administering tamoxifen (TM) at E8.5. Shh: Sonic Hedgehog; YFP: yellow fluorescent protein. CreER: gene encoding a CRE-Estrogen Receptor fusion protein. (B) Schematic of horizontal organotypic slice culture preparations. Green regions represent location

*Figure 4 continued on next page*

Figure 4 continued

of mDA neurons in the embryonic brains (left) and in horizontal slices (right). Red dashed line indicates dorsoventral level of slices. Black arrows indicate direction of tangential mDA migration. (C) Maximum intensity projection of an image of a control slice at t = 0. Orange rectangle indicates location of cell shown in E. (D) Trajectories of tracked neurons in slice shown in (C) after imaging for 270 min. Trajectory in orange represents trajectory of neuron in (E). (E) Maximum intensity projection frames of time-lapse imaging show soma and processes of a tangentially migrating cell. Track overlay frames show the position of the soma (green circle) and trajectory of the cell (orange line) analyzed with the semi-automated tracking plugin TrackMate in Fiji. Magenta dots and circles represent tracked soma of close-by cells at different z-levels. (F) Speed profile of cell in (E) shows large variations in speed over time, with a maximum speed (dashed arrow) that is much higher than the average speed (dashed horizontal line). Rest phase (soma speed less than 10 µm/hr) is indicated in red. Scale bars: (C,D) 50 µm, (E) 20 µm.

The online version of this article includes the following figure supplement(s) for figure 4:

**Figure supplement 1.** Average speed distributions and speed profiles of SN-mDA neurons.

studies have shown that the effect of Reelin varies depending on the brain region and type of neuron analyzed (*Simó et al., 2010*; *Britto et al., 2014*; *Britto et al., 2011*; *Wang et al., 2018*). We have previously demonstrated that inhibiting Reelin in ex vivo slices results in a decrease in average speed of SN-mDA neurons over long periods of imaging (*Bodea et al., 2014*). In our current analysis we found no significant difference in the overall distribution of average speeds of the SN-mDA population in *Dab1*[-/-] slices compared to control slices (*Figure 5A*). However, overall distribution of max-speeds was significantly shifted towards lower speeds in the absence of Reelin signaling (control: 25th percentile = 12.4 µm/hr, median = 23.6 µm/hr, 75th percentile = 48.1 µm/hr, maximum = 183 µm/hr; *Dab1*[-/-]: 25th percentile = 10.1 µm/hr, median = 15 µm/hr, 75th percentile = 29.8 µm/hr, maximum = 133.7 µm/hr; *Figure 5B*).

We then asked whether this shift towards lower max-speeds in *Dab1*[-/-] SN-mDA neurons was accompanied by other changes in migratory behavior, or whether the neurons simply displayed lower max-speeds while maintaining the same migratory, directional and morphological characteristics as control SN-mDA neurons. For this analysis, we divided control and *Dab1*[-/-] neurons into four groups based on the lower and upper quartiles of the *Dab1*[-/-] max-speed distribution. We defined these groups in the following manner: non-migratory cells with max-speeds of less than 10 µm/hr (control = 126/806, *Dab1*[-/-] = 205/844), 'slow' cells with max-speeds from 10 to 30 µm/hr (control = 355/806, *Dab1*[-/-] = 430/844), 'moderate' cells with max-speeds from 30 to 60 µm/hr (control = 186/806, *Dab1*[-/-] = 139/844) and fast' cells with max-speeds > 60 µm/hr, control = 139/806, *Dab1*[-/-] = 70/844) (*Figure 5B*). Non-migratory cells failed to move more than 1.7 µm in any two consecutive frames of analysis and were not included into the further analysis. Using this classification, a lower percentage of SN-mDA neurons reached moderate or fast migration speeds in *Dab1*[-/-] slices compared to controls, while the proportion of both non-migratory and 'slow' cells was increased (*Figure 5B*).

Next, we asked how frequently migrating SN-mDA neurons moved with soma speeds comparable to their max-speeds and whether the fraction of total time-points spent in high migratory speeds was different in control and *Dab1*[-/-] populations. To evaluate this, we used the criteria previously defined for max-speeds, but applied them to individual soma speeds for each cell at each time point. For example, we analyzed the fraction of time (percentage of total time-points) spent by each fast' cell with a soma speed of more than 60 µm/hr (fast migratory phase), 30–60 µm/hr (moderate migratory phase), 10–30 µm/hr (slow migratory phase) and less than 10 µm/hr (resting phase). In control slices, 'fast', 'moderate' and 'slow' cells spent a predominant fraction of time at rest (62.6 ± 20%; 68.5 ± 18.2%, 85.7 ± 11.1%, respectively) and were frequently in a slow migratory phase (26.8 ± 17.4%, 25.1 ± 16.3%, 14.2 ± 11.1%, respectively). 'Fast' and 'moderate' cells achieved the moderate migratory

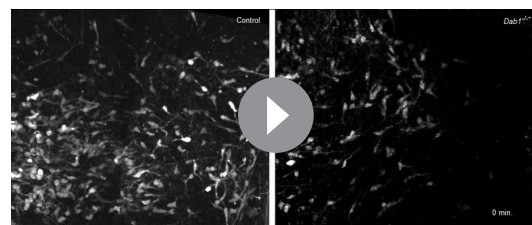

**Video 1.** Time lapse imaging with two photon excitation of ex vivo embryonic slices of the ventral midbrain. Time-lapse imaging of control (left) and *Dab1*[-/-] (right) organotypic slices with mosaic labelling of SN-mDA neurons reveals aberrant orientation and slower migration of *Dab1*[-/-] mDA neurons.
https://elifesciences.org/articles/41623#video1

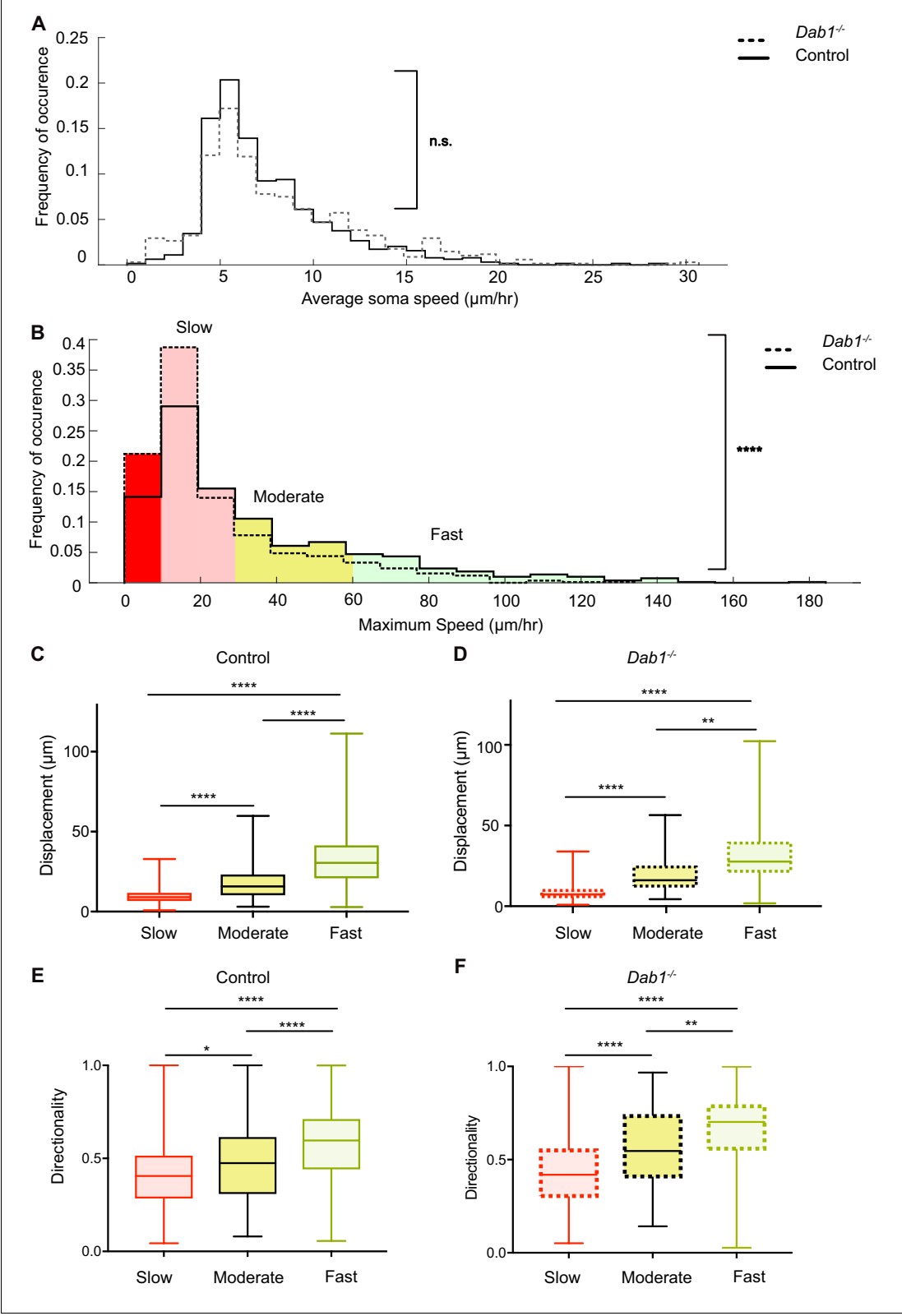

**Figure 5.** Reelin promotes infrequent, fast movements in mDA neurons. (**A**) The distribution of average speeds for the entire population of tracked control and *Dab1^-/-* mDA neurons. Overall distribution of average soma speeds of mDA neurons is not significantly altered in *Dab1^-/-* slices compared to controls. Note that entire distribution of control (solid black line) and *Dab1^-/-* mDA neurons (dashed black line) were compared (p=0.0657, Mann-Whitney's test, n = 806 control, 844 *Dab1^-/-* mDA neurons). (**B**) The same analysis as shown in (**A**) was carried out for the distributions of maximum (max)

*Figure 5 continued on next page*

*Figure 5 continued*

speed. At the population level (distributions were compared), max-speeds of *Dab1*$^{-/-}$ mDA neurons were shifted significantly towards slower speeds compared to controls (p<0.0001, Mann-Whitney's test, n = 806 control, 844 *Dab1*$^{-/-}$ mDA neurons). Non-migratory (max-speed 0–10 µm/hr), slow (30–10 µm/hr), moderate (60–30 µm/hr) and fast cells (>60 µm/hr) are indicated by dark red, light red, yellow and light green background colors, respectively. (C,D) Total displacement (3D) of mDA neurons is significantly higher in moderate compared to slow mDA neurons, and higher in fast mDA neurons compared to moderate neurons in both control (C) and *Dab1*$^{-/-}$ (D) brains (****p<0.0001, **p<0.01, Kruskal-Wallis test; n = 680 control; n = 639 *Dab1*$^{-/-}$ cells, three slices/genotype). (E,F) Directionality (defined as ratio of total displacement to path length) in control and *Dab1*$^{-/-}$ slices is the least in slow mDA neurons, higher in moderate mDA neurons and the highest in fast mDA neurons (*p<0.05, **p<0.01, ****p<0.0001, Kruskal-Wallis test; n = 680 control; n = 639 *Dab1*$^{-/-}$ cells, three slices/genotype).

The online version of this article includes the following figure supplement(s) for figure 5:

**Figure supplement 1.** Variation in instantaneous soma speed of mDA neurons.

**Figure supplement 2.** Individual fast, moderate and slow mDA neurons from *Dab1*$^{-/-}$ slices have similar directionality and displacement profiles as mDA neurons in control slices.

phase in only a few frames (5.5 ± 5.5% and 6.3 ± 3.9, respectively), and the fast migratory phase (only in 'fast' cells) was equally infrequent (5.5 ± 2.2%) (*Figure 5—figure supplement 1*). The amount of time SN-mDA neurons of the same max-speed group spent in the resting phase or in the respective migratory phases was comparable between individual cells in control and *Dab1*$^{-/-}$ slices (*Figure 5—figure supplement 1D–F*).

In summary, these results demonstrate that SN-mDA migration has two distinct modes: a frequent slow migration phase seen in all migrating SN-mDA neurons and an infrequent moderate-to-fast phase occurring in a subset of SN-mDA neurons. These phases are superimposed over frequent periods of rest. Reelin signaling increases the proportion of migratory mDA neurons and the likelihood of moderate-to-fast movements in migrating mDA neurons. As moderate-to-fast migratory phases are only attained in very few frames in our experiments, the average speed distribution of SN-mDA neurons are however not changed in *Dab1*$^{-/-}$ slices compared to control slices.

## The Reelin-promoted infrequent fast movements of mDA neurons contribute to large directed cell displacements

We next asked whether max-speeds and directionality of migration were linked. We computed directionality as the ratio of total displacement (the 3D displacement between the initial and final positions of the neurons) to path length (the distance travelled by each neuron summed up irrespective of direction; *Petrie et al., 2009*) for migrating SN-mDA populations in control and *Dab1*$^{-/-}$ slices. A high value of directionality (maximum value = 1) indicates almost no change in migratory direction while low values indicate frequent changes in direction. We found that directionality as well as total displacement generally increased with increasing max-speeds in SN-mDA populations from both control and *Dab1*$^{-/-}$ slices (*Figure 5C–F*; *Figure 5—figure supplement 2*). These data indicate that the infrequent moderate-to-fast movements in SN-mDA neurons result in major contributions to the directed migration of these cells. Since Reelin signaling increases the fraction of SN-mDA neurons that are able to undergo moderate-to-fast movements, Reelin supports directed migration of mDA neurons on a population level.

## Reelin promotes preference for laterally-directed migration in mDA neurons

As tangential migration ultimately results in SN-mDA migration away from the midline, we analyzed the trajectories of migratory SN-mDA neurons in the presence and absence of Reelin signaling. We determined the 'trajectory angle' for each cell as the angle between the midline (y-axis in live-images) and the cell's displacement vector (*Figure 6A*). Thus, a trajectory angle of 90° indicates a cell whose total movement is precisely aligned to the lateral axis (x-axis in live-images). We defined a cell as migrating laterally if its trajectory angle was between 45–135°. We then evaluated the angular mean and standard deviation ($\sigma_{ang}$) for SN-mDA populations in control and *Dab1*$^{-/-}$ slices (*Berens, 2009*). We found that SN-mDA neurons from control slices displayed an anisotropy towards lateral migratory directions (mean 92.5°, $\sigma_{ang}$ 68.4) while *Dab1*$^{-/-}$ SN-mDA neurons showed a significantly reduced preference for lateral migration (mean 27.5°, $\sigma_{ang}$ 70.4) (*Figure 6B–D*; see materials and methods for analysis of circular variables).

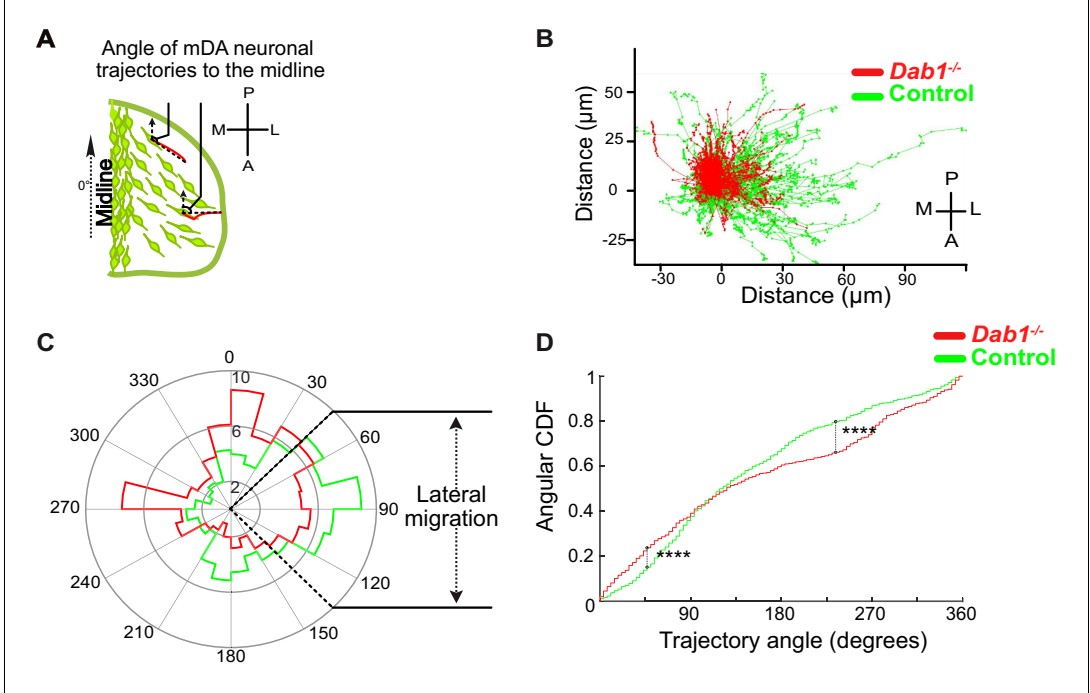

**Figure 6.** Lateral migration of mDA neurons is reduced in the absence of Reelin signaling. (**A**) Schematic showing trajectory angle measurement. (**B**) Trajectories of control (green) and *Dab1⁻/⁻* (red) mDA neurons (from 1 control and 1 *Dab1⁻/⁻* slice imaged over the same duration), plotted relative to their starting point show loss of lateral directionality in *Dab1⁻/⁻* mDA neuron trajectories. (**C**) Polar histogram for angle of the mDA trajectories to the midline (0°) for all control (green) and *Dab1⁻/⁻* (red) mDA neurons analyzed. (**D**) Circular statistical analysis for angular distributions in (**C**) of control (green) and *Dab1⁻/⁻* (red) mDA neurons shows significant decrease in lateral anisotropy for *Dab1⁻/⁻* slices (****p<0.0001, Kuiper's test for circular variables; n = 680 control, n = 639 *Dab1⁻/⁻* mDA neurons). CDF: cumulative distribution function.

Next, to evaluate if 'fast', 'moderate' and 'slow' cell populations of control and *Dab1⁻/⁻* slices showed differences in their preference for lateral migration, we analyzed their trajectories separately. We found that trajectories of all three SN-mDA groups were anisotropic in controls, favoring migration towards lateral directions, but this anisotropy was greater in 'fast' and 'moderate' cells than in 'slow' cells (*Figure 7A,D,G*). Resolving this further into individual slow, moderate and fast migratory phases in the migratory mDA population, we also found that individual moderate-to-fast phases were more anisotropic than slow phases (*Figure 7—figure supplement 1A,C,E*).

In the absence of Reelin signaling, the trajectory profiles of 'slow' neurons were significantly altered with a complete loss of anisotropy towards lateral directions (mean −12.3°, $\sigma_{ang}$ 69.7°) (*Figure 7A–C*). In contrast, 'fast' and 'moderate' neurons still navigated to more lateral regions in *Dab1⁻/⁻* slices and their trajectory angle distributions were nearly identical to control neurons (*Dab1⁻/⁻* 'fast' neurons: mean 81°, $\sigma_{ang}$ 57.9°; 'moderate' neurons: mean 69.4°, $\sigma_{ang}$ 58.7°) (*Figure 7D–I*). This finding also applies to slow, moderate and fast phases: slow phases are weakly laterally-directed in controls, but in the absence of Reelin signaling individual slow migratory movements lose their slight lateral preference (*Figure 7—figure supplement 1B,D,F* and data not shown). These results show that Reelin signaling promotes lateral migration of SN-mDA neurons by increasing the fraction of SN-mDA neurons undergoing moderate-to-fast movements that are strongly biased for tangential movements and by promoting lateral anisotropy of slowly migrating neurons.

## mDA neurons are predominantly associated with bipolar morphology during moderate-to-fast phases of migration

Having thus defined the complex regulation of SN-mDA speed and trajectory profiles by Reelin signaling, we investigated the cellular morphology that underlies mDA tangential migration. Since the dynamic cell morphologies of migrating SN-mDA neurons have not been assessed previously, we first evaluated morphological changes in control SN-mDA neurons. Some cells had a stable,

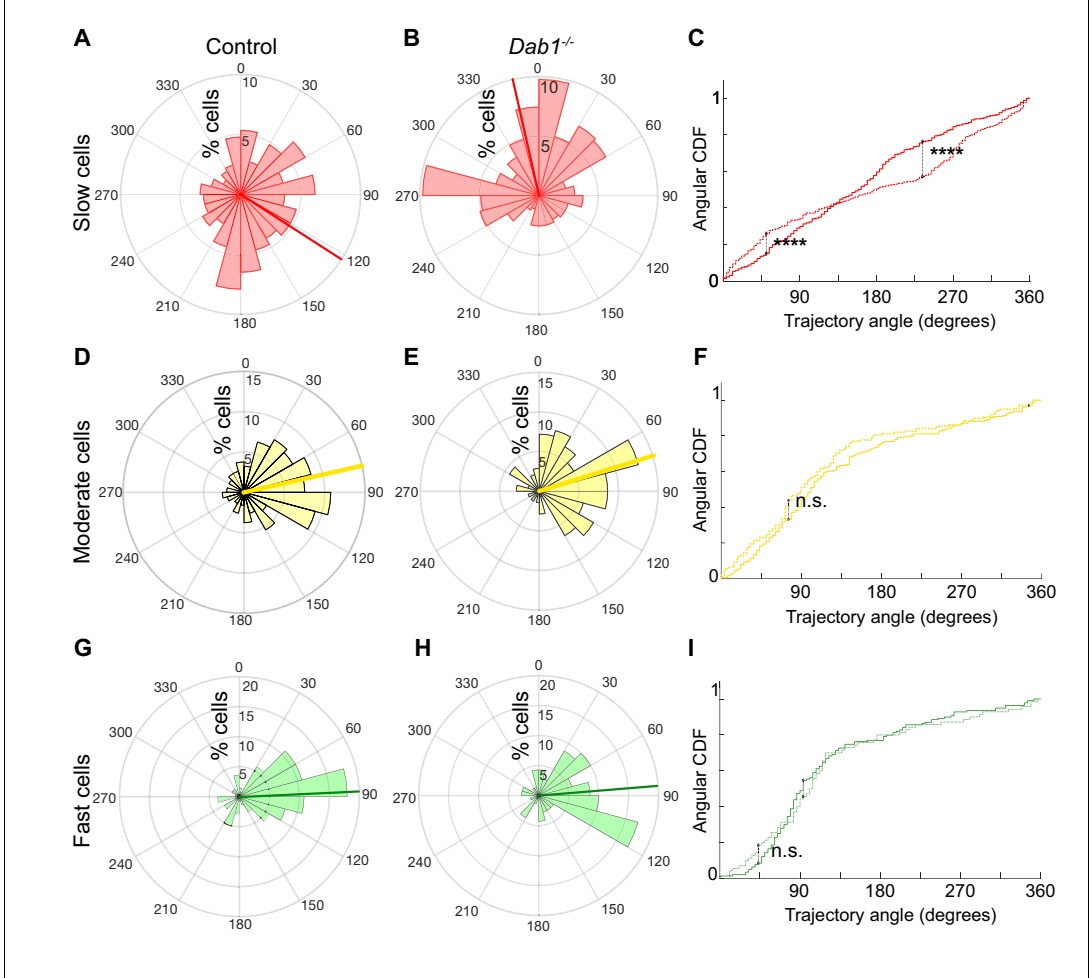

**Figure 7.** Reelin promotes preference for lateral migration in slow mDA neurons. Polar histogram for the angle of mDA trajectories to the midline. Radial axes represent the percentage of cells that migrate with various angles to the midline. (A,B) shows that slow cells have the least preference for lateral migratory direction in both control (A) and *Dab1*[-/-] (B) slices. (C) Circular statistical analysis for angular distributions of slow mDA neurons shows significant loss of preference for lateral migration in slow *Dab1*[-/-] mDA neurons compared to controls (****p<0.0001, Kuiper's test for circular variables; n = 355 control, 480 *Dab1*[-/-] mDA neurons). (D–I) Moderate (D,E) and fast mDA neurons (G,H) show high preference for lateral migration. Moderate (F; n = 186 control, n = 139 *Dab1*[-/-] mDA neurons) and fast mDA neurons (I; n = 139 control, n = 70 *Dab1*[-/-] mDA neurons) are laterally directed and their angular cumulative distribution functions (CDF) are comparable in control and *Dab1*[-/-] slices. Red (A,B), yellow (D,E) and green (G,H) lines represent mean angular direction for slow, moderate and fast populations, respectively.

The online version of this article includes the following figure supplement(s) for figure 7:

**Figure supplement 1.** Lateral migration occurs predominantly during moderate and fast migratory phases of mDA neurons.

unbranched leading process (LP), and did not change their morphology, while other cells displayed dynamic LPs, that extended, retracted and branched frequently over time (*Figure 8A–D*; *Figure 8— figure supplement 1*; *Video 2*).

We studied the cell morphology of SN-mDA neurons (70 'fast', 40 moderate' and 40 'slow' cells) in control and in *Dab1*[-/-] slices (49 'fast', 40 'moderate' and 40 'slow' cells) and examined whether slow, moderate and fast migratory phases were associated with specific morphologies (for details of morphological analysis see materials and methods). We defined three morphological categories: a neuron was considered to be 'bipolar-unbranched' when a maximum of two processes arose directly from the soma and the LP was unbranched. Bipolar cells that extended a branched LP were defined as 'bipolar-branched'. Neurons with more than two processes arising from the soma were defined as 'multipolar' (*Figure 8A,C*; *Figure 8—figure supplement 1*). The morphology of SN-mDA neurons

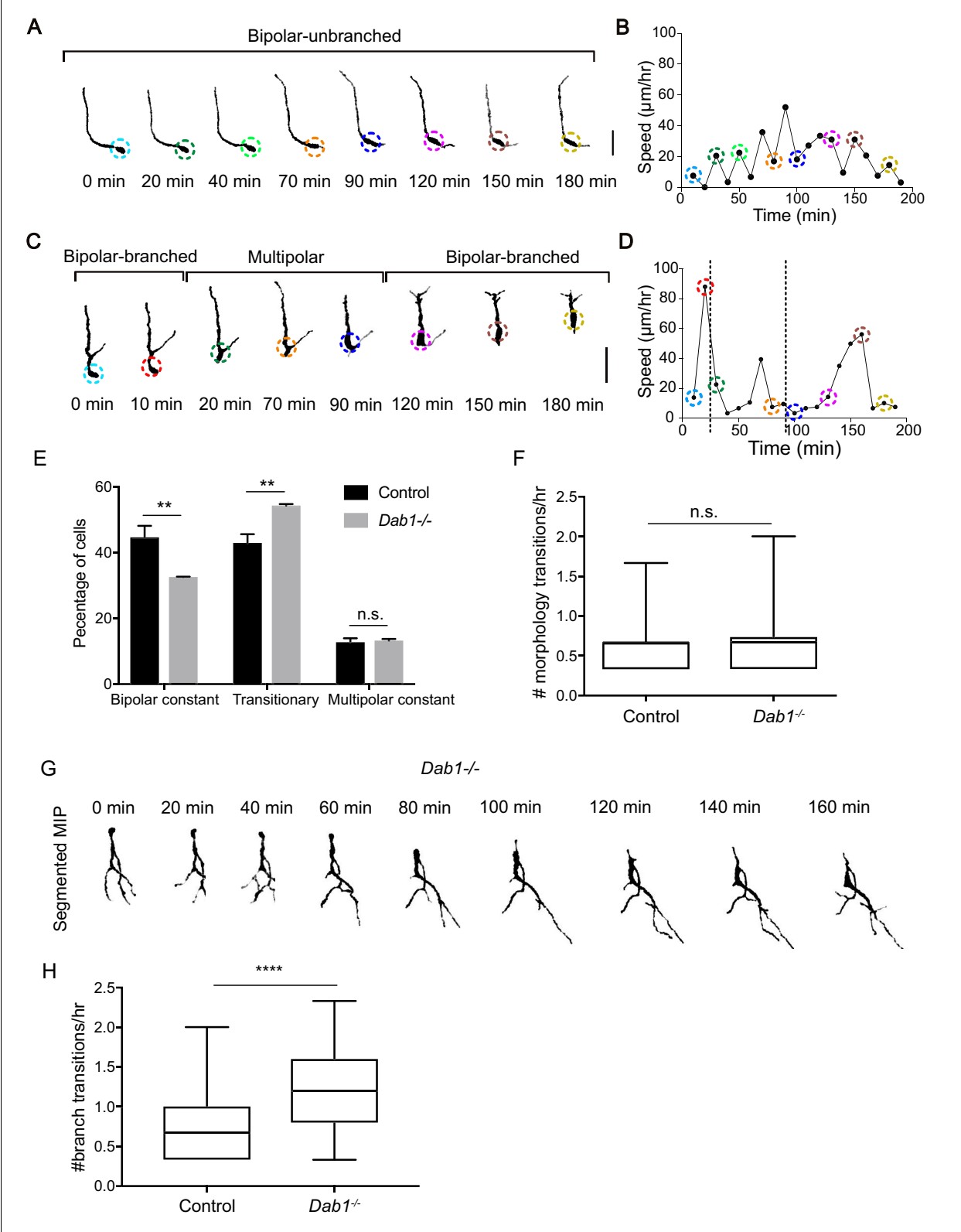

**Figure 8.** Reelin promotes stable morphology of migrating SN-mDA neurons. (**A**) Control mDA neuron displaying a 'bipolar-unbranched' morphology at all analyzed time-points. (**B**) Soma speed profile of mDA neuron shown in (**A**). (**C**) Control mDA neuron transitioning between bipolar and multipolar morphology. At t = 0 min, the cell has a branched leading process (LP), the cell soma moves along the LP to reach the branch-point and takes up a multipolar morphology (t = 20 min). The cell remains multipolar until t = 90 min, after which one process is retracted (t = 120 min) and the cell resumes

*Figure 8 continued on next page*

Figure 8 continued

a bipolar morphology (t = 150 min). Bipolar phase: one or two processes arise directly from the soma. Multipolar phase: more than two processes arise directly from the soma. (A,C) Colored circles: soma as defined by the tracking process. Scale bar: 25 μm. (D) Soma speed for the neuron in (C) is higher during its bipolar phase. (E) Relative proportion of constantly bipolar mDA neurons are decreased, while transitionary mDA neurons are increased in $Dab1^{-/-}$ slices. (F) Frequency of transitions from multipolar to bipolar phase (and vice versa) are not significantly altered in the absence of Reelin signalling (p=0.6922; Mann-Whitney's test). (G) Bipolar and multipolar phase of a $Dab1^{-/-}$ transitionary mDA neuron. In this example, the bipolar phase lasts from t = 0 min to t = 60 min. In the multipolar phase (starting at t = 80 min) many unstable protrusions form. Scale bar: 25 μm. See **Figure 8—figure supplement 3** for more detail. (H) Quantification of appearance and disappearance of branches (defined as branch transitions per hour) in control and $Dab1^{-/-}$ mDA transitionary neurons shows a significant increase in branch transitions in mDA neurons in $Dab1^{-/-}$ slices (****p<0.0001; Mann-Whitney's test).

The online version of this article includes the following figure supplement(s) for figure 8:

**Figure supplement 1.** Morphological characterization of mDA neurons in control slices.

**Figure supplement 2.** mDA neurons are predominantly associated with bipolar morphology during moderate-to-fast phases of migration.

**Figure supplement 3.** Absence of Reelin signaling results in the formation of unstable protrusions on the soma and leading process of mDA neurons.

**Figure supplement 4.** Greater spread in length of leading process in $Dab1^{-/-}$ mDA neurons.

evaluated based on YFP expression was indistinguishable from their morphology as assessed by TH-immunostaining in cleared whole-mount brains at E13.5 (*Video 3*).

To investigate whether specific morphologies observed in SN-mDA neurons were associated with specific migratory speeds, we broke down the morphology of these cells into time points during which they were in bipolar-unbranched, bipolar-branched or multipolar phases and paired their morphology with soma speed (as calculated by change in soma position between the current and the subsequent time point) (*Figure 8A–D*). Bipolarity was predominant in all phases of migration, but in both control and $Dab1^{-/-}$ SN-mDA neurons, fast and moderate migratory phases were almost exclusively associated with bipolar morphology. In contrast, about a third of slow migratory phases were associated with multipolar morphology (*Figure 8—figure supplement 2*). Hence, while slow migratory phases can occur in either bipolar or multipolar morphology, fast and moderate migration events are predominantly associated with bipolar morphology.

## mDA neurons display unstable branch and leading process morphology in the absence of Reelin signaling

In time-lapse data-sets, some mDA neurons transitioned between bipolar and multipolar morphology, while others maintained either a bipolar or multipolar morphology during imaging. We next examined the proportions of migrating SN-mDA neurons that displayed a constant bipolar (branched and unbranched), constant multipolar or transitionary morphology over time (*Figure 8A, C*; *Figure 8—figure supplement 1*). This analysis enabled us to ask whether morphological stability is altered in the absence of Reelin signaling. In controls, transitionary cells made up about 40% of the total population. The proportion of transitionary cells was significantly increased in the $Dab1^{-/-}$ population, while the population of bipolar neurons was decreased (*Figure 8E*, *Table 1*). Within the transitionary population, we found however no difference in the frequency of transitions between bipolar and multipolar morphologies for each neuron (defined as number of morphology transitions per hour) in $Dab1^{-/-}$ compared to control slices (*Figure 8F*). We then examined the appearance and disappearance of processes both on the soma and the LP of transitionary neurons (n = 64 in control, n = 70 in $Dab1^{-/-}$) in further detail (*Figure 8G*, *Figure 8—figure supplement 3*). We found that these branch transition events were significantly more frequent in $Dab1^{-/-}$ SN-mDA transitionary

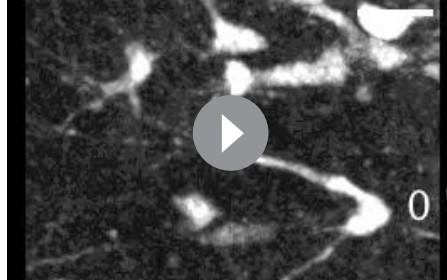

**Video 2.** SN-mDA neurons display dynamic cell morphology. 3D projection of a transitionary mDA neurons at t = 0 min (360° rotation) followed by MIP frames of the same neuron at subsequent time-points. Migratory spurts only occur in bipolar morphology while cell remains stationary or displays slow migration during multipolar phase.
https://elifesciences.org/articles/41623#video2

**Video 3.** Morphology as detected by YFP mosaic labelling is similar to morphology detected by TH antibody. Example SN-mDA neuron from fixed, cleared whole-mount embryonic brain of the same age as used in time-lapse experiments (E14.5) shows similar morphology with YFP (green) and TH (magenta) immunostaining.
https://elifesciences.org/articles/41623#video3

neurons (*Figure 8H*), since *Dab1⁻/⁻* neurons displayed short, transient protrusions that appeared on the soma and LP for only a few time frames before disappearing (*Figure 8—figure supplement 3*).

Finally, we randomly selected 20 control and 20 *Dab1-/-* mDA neurons with maximum soma speed of more than 10 µm/hr and manually traced their morphology in 3D for the first 19 imaging time-points (*Figure 8—figure supplement 4*). In all control and *Dab1⁻/⁻* mDA neurons, the LP remained stable and visible during the duration of imaging. We then compared the length of the LP (plus cell body) in control and *Dab1⁻/⁻* mDA neurons and found that mDA neurons in *Dab1⁻/⁻* slices displayed a broader distribution of LP length with very long and very short LPs (*Figure 8—figure supplement 4G*). Hence, in the absence of Reelin signaling, SN-mDA neurons display aberrant changes in morphology characterized by an increased proportion of transitionary neurons, an increase in unstable processes on the cell soma and LP and a greater variation in LP length.

## Reelin downstream signaling in the ventral midbrain

As it is not known which downstream components of the Reelin signaling pathway regulate SN-mDA tangential migration, we investigated Reelin signaling events that were previously shown to influence neuronal polarity in migrating neurons in the cortex, hippocampus or spinal cord. Reelin signaling leads to the activation (phosphorylation) of PI3K (Phosphatidylinositol-4,5-bisphosphate 3-kinase) through DAB1. PI3K activation results in phosphorylation (activation) of LIMK1 (Lim domain kinase 1) via Rac1/Cdc42 and PAK1. P-LIMK1 inactivates (phosphorylates) Cofilin1, an actin depolymerizing protein of the ADF/Cofilin family. Reelin-mediated inactivation of Cofilin 1 ultimately leads to the stabilization of the actin cytoskeleton and has been implicated in stabilizing LPs of radially migrating cortical neurons as well as in preventing the aberrant tangential migration of neurons of the autonomous nervous system in the spinal cord (*Maciver and Hussey, 2002*; *Krüger et al., 2010*; *Chai et al., 2009*; *Franco et al., 2011*; *Frotscher et al., 2017*). To detect a potential misregulation of these downstream events in absence of Reelin signaling, we performed immunoblotting on E14.5 embryonic ventral midbrain tissue for p-LIMK1/LIMK1 and p-Cofilin1/Cofilin1. We did not detect significant differences in protein levels or in relative phosphorylation levels (*Figure 9* and data not shown). Hence, we conclude that the regulation of LIMK1/Cofilin1 activity is unlikely to be the key event in controlling cytoskeletal stability in migrating mDA neurons downstream of Reelin signaling.

**Table 1.** Morphology of mDA neurons in control and *Dab1-/-* slices.

| Cell type | Morphology | Control | | | Dab1-/- | | |
|-----------|------------|---------|---------|---------|---------|---------|---------|
| | | Slice 1 | Slice 2 | Slice 3 | Slice 1 | Slice 2 | Slice 3 |
| | Bipolar | 15/27 | 12/26 | 7/17 | 5/16 | 6/15 | 5/18 |
| Fast | Transitionary | 12/27 | 13/26 | 10/17 | 9/16 | 8/15 | 12/18 |
| | Multipolar | 0/27 | 1/26 | 0/17 | 2/16 | 1/15 | 1/18 |
| | Bipolar | 8/16 | 4/10 | 4/14 | 4/13 | 5/19 | 5/8 |
| Moderate | Transitionary | 7/16 | 6/10 | 10/14 | 9/13 | 14/19 | 3/8 |
| | Multipolar | 1/16 | 0/10 | 0/14 | 0/13 | 0/19 | 0/8 |
| | Bipolar | 3/11 | 7/12 | 7/17 | 5/14 | 5/15 | 2/11 |
| Slow | Transitionary | 2/11 | 1/12 | 3/17 | 5/14 | 5/15 | 5/11 |
| | Multipolar | 6/11 | 4/12 | 7/17 | 4/14 | 5/15 | 4/11 |

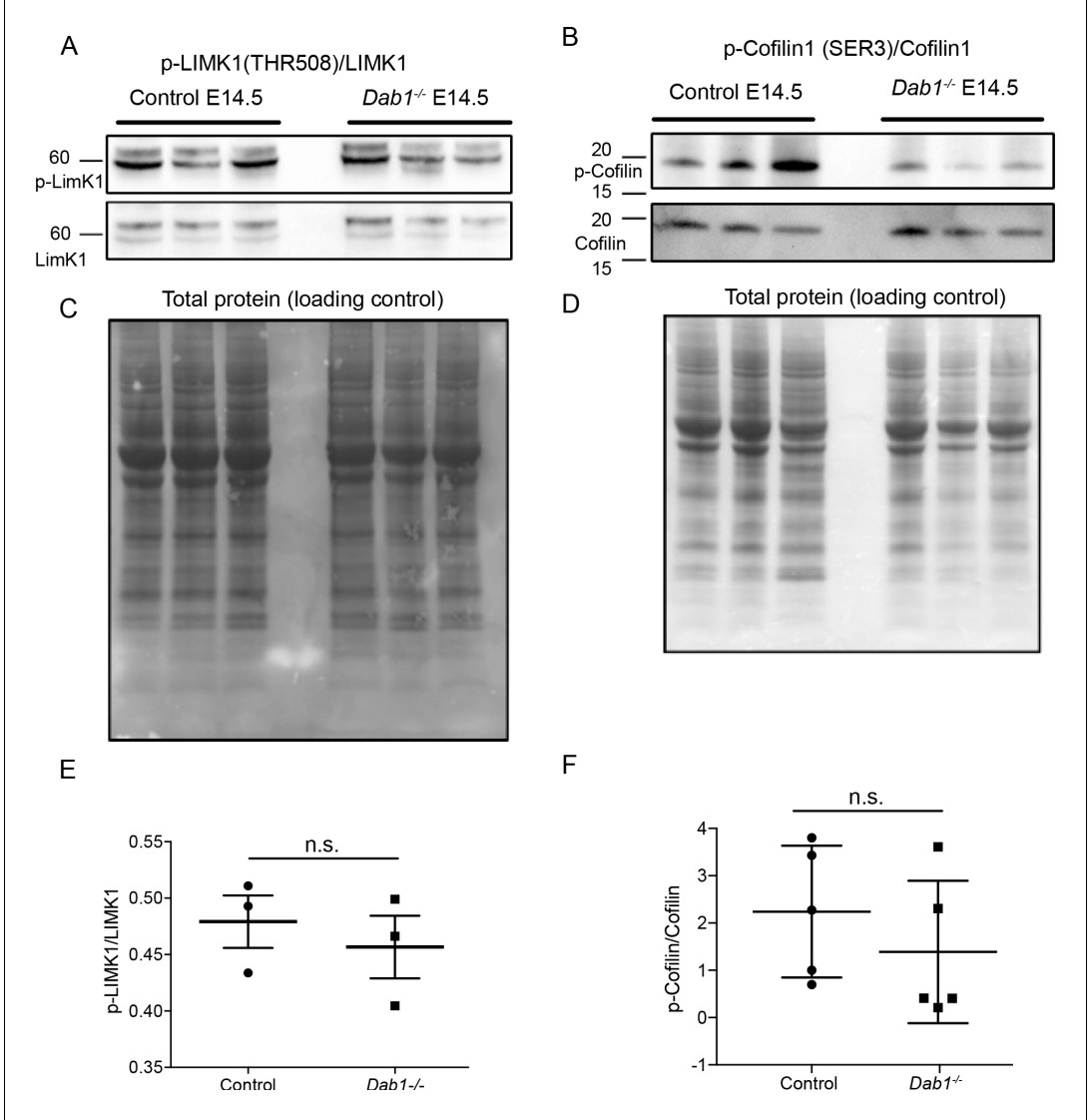

**Figure 9.** Phosphorylation levels of proteins in the canonical Reelin signaling pathway are not altered in $Dab1^{-/-}$ ventral midbrain at E14.5. (A,E) Relative phosphorylation levels of LIMK1 are not significantly altered in $Dab1^{-/-}$ ventral midbrains. p-value=0.5682, Student's t-test (n = 5 brains/genotype). (B,F) Relative phosphorylation levels of Cofilin1 are not significantly changed in $Dab1^{-/-}$ ventral midbrains. p-value=0.854, Student's t-test (n = 5 brains/ genotype). (C,D) Total protein in the membrane, stained with Amido black, was used as loading control.

The online version of this article includes the following figure supplement(s) for figure 9:

**Figure supplement 1.** CDH2 expression levels in the ventral midbrain of control and $Dab1^{-/-}$ brains at E14.5.

**Figure supplement 2.** Organization of microtubules is not altered in mDA neurons in the absence of Reelin signaling.

Next, we examined Cadherin2 (CDH2) expression in the ventral midbrain. Reelin signaling controls somal translocation of radially migrating cortical neurons by modulating cell adhesion properties through regulation of CDH2 via the Crk/C3G/Rap1 pathway (*Jossin and Cooper, 2011*; *Gärtner et al., 2012*; *Gil-Sanz et al., 2013*; *Cooper, 2014*; *Matsunaga et al., 2017*). Relative protein levels of CDH2 were similar in tissue lysates from control and $Dab1^{-/-}$ E14.5 ventral midbrain (*Figure 9—figure supplement 1*). Whether CDH2 levels are altered at the membrane of mDA neurons in $Dab1^{-/-}$ mice could not be assessed, since the immunostaining for CDH2 on sections was not of sufficient quality to make a clear assessment of changes in membrane localization.

In the cortex, Reelin positively regulates microtubule dynamics in cortical neurons during development (*Meseke et al., 2013*). To assess whether the organization of the microtubule cytoskeleton might be altered in mDA neurons in absence of Reelin signaling, we analyzed the relative abundance

of stable and instable microtubules in the ventral midbrain of control and $Dab1^{-/-}$ embryos by immunostaining for α-tubulin, acetylated α-tubulin and end binding protein 3 (EB3). EB3 is a component of a large protein complex (*van de Willige et al., 2016*) that regulates growth of microtubules at their plus end and is an established marker to examine the dynamics of microtubule growth in neurons (*Stepanova et al., 2003*). The acetylation of tubulin contributes to the stabilization of tubulin (*Fernández-Barrera and Alonso, 2018*). In contrast to what has been reported in the cortex, we could however not detect an obvious change in the distribution or expression level of these tubulin markers in the ventral midbrain of $Dab1^{-/-}$embryos (*Figure 9—figure supplement 2*), indicating that the loss of *Dab1* has likely no major effect on the microtubule cytoskeleton.

## Discussion

The correct tangential migration of mDA neurons is crucial for the formation of the SN. Our study provides the first comprehensive insight into speed, trajectory and morphology profiles of tangentially migrating mDA neurons, and uncovers the alterations of tangential migratory behavior that result in aberrant SN formation in the absence of Reelin signaling (*Figure 10*).

### Reelin signaling directly regulates tangential migration of SN-mDA neurons

A number of previous studies established the importance of Reelin in the formation of the SN (*Nishikawa et al., 2003*; *Kang et al., 2010*; *Sharaf et al., 2013*; *Bodea et al., 2014*), but it remained to be elucidated whether Reelin is directly required for the tangential migration of SN-mDA neurons. Studies in cortex have shown that while Reelin is directly required for the stabilization of the LP and for the orientation of radially-migrating cortical projection neurons (*Franco et al., 2011*), Reelin also indirectly affects migration through regulating radial glia cell process extension, morphology and maturation (*Hartfuss et al., 2003*; *Keilani and Sugaya, 2008*). Tangentially migrating cortical interneurons are only indirectly affected by Reelin signaling: the improper cortical layering caused by defective radial migration in absence of Reelin signaling ultimately results in mispositioning of interneurons (*Yabut et al., 2007*). Reelin also plays a role in interneuron precursors that undergo tangential chain migration to the olfactory bulb. However, it does not modulate tangential migration directly but rather acts as a detachment signal that regulates the switch form tangential chain migration to radial migration (*Hack et al., 2002*). Evidence for a direct function of Reelin signaling in tangential neuronal migration comes from sympathetic preganglionic neurons in the spinal cord. In these neurons, Reelin has been shown to stabilize LPs via the phosphorylation of during tangential migration thereby preventing aberrant migration (*Phelps et al., 2002*; *Krüger et al., 2010*).

To explore whether Reelin has a direct role in tangential migration of SN-mDA neurons, we inactivated *Dab1* in SN-mDA neurons starting at the onset of their tangential migration without affecting their earlier radial migration step and without inactivating *Dab1* in other cell populations in the ventral midbrain. The similarity in mediolateral distribution of SN-mDA neurons in $Dab1^{-/-}$ and in *Dab1* CKO implies that Reelin signaling has a direct effect on migrating SN-mDA neurons. We also confirmed that the GIRK2-expressing mDA population, which consists of lateral VTA- and SN-mDA neurons was distributed in a similar manner than what we reported previously for $Dab1^{-/-}$ mice (*Bodea et al., 2014*). Investigation of additional markers that label SN-mDA neurons more specifically, such as *Lmo3* and SOX6, showed that the medially misplaced SN-mDA neurons were partially intermingled with VTA-mDA neurons, but only at the border between SN and lateral VTA. These results imply that in absence of Reelin signaling in mDA neurons, the separation of SN- and VTA-mDA neurons is not fully completed and SN-mDA neurons lose their ability to undergo the long-range tangential migration necessary to form the laterally-positioned SN. Thus, our findings are the first demonstration of Reelin as a direct regulator of tangential neuronal migration in the brain.

Interestingly, the inactivation of *Dab1* in mDA neurons and the disorganization of the SN and VTA did not obviously alter the density of projections to the striatum. This indicates that the axons of mislocalized mDA neurons may still find their way and correct position in the medial forebrain bundle and then rely on intrinsic markers (cell adhesion molecules) to navigate to their particular projection target (*Brignani and Pasterkamp, 2017*). However, deeper insight into the consequences of the migratory phenotype on the connectivity of mDA neurons would require further experiments

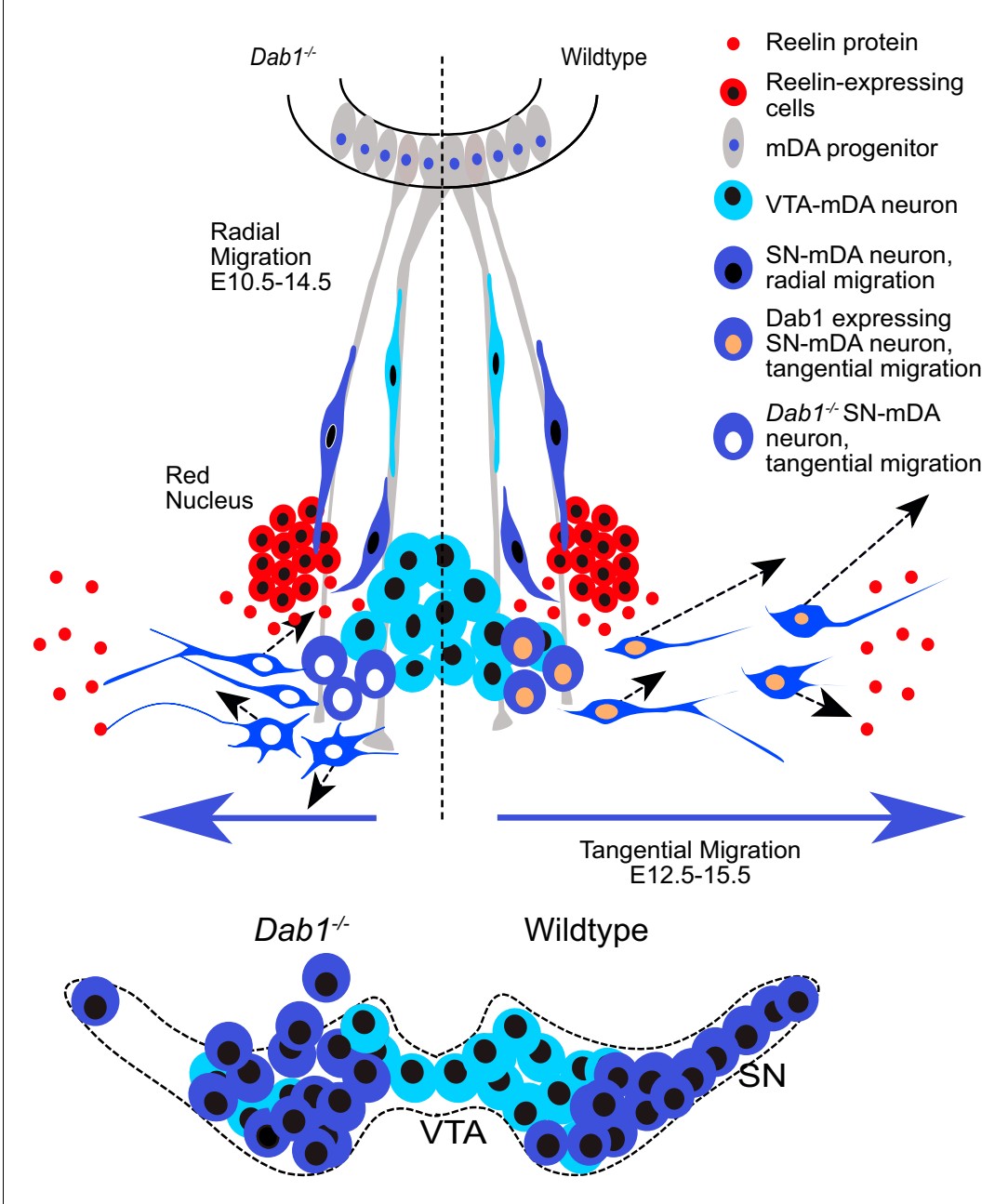

**Figure 10.** Schematic representation of Reelin regulation in mDA tangential migration. Reelin directly regulates lateral, tangential migration of mDA neurons and promotes fast, laterally directed mode of migration by regulating small lateral movements and stabilizing morphology of mDA neurons. In absence of Reelin signaling, slow movements in mDA neurons lose their lateral anisotropy, morphologies are less stable during migration and the fraction of neurons undergoing fast, laterally directed migration is reduced. This results in a medial clustering of SN-mDA neurons at late embryonic and postnatal stages.

such as specific tracing of projections from SN neurons or single cell labeling to visualize individual axonal arbors.

## Reelin protein is localized in the lateral ventral midbrain

In the ventral midbrain, *Reelin* mRNA is restricted to the cells of the red nucleus at E13.5 and E14.5 (*Bodea et al., 2014*; *Figure 3*). Using immunostaining, we show that Reelin protein is distributed more broadly at these stages. Strong labeling is seen in regions lateral to the migrating SN-mDA,

while weaker staining is observed in the area where SN-mDA neurons are localized. Thus, the Reelin protein distribution that we describe here is consistent with a direct role of Reelin in regulating SN-mDA migration. Whether the red nucleus is the only source for Reelin in the ventral midbrain or whether there are additional sources remains to be investigated. Mouse mutants in which the RN is only partially formed do not show any obvious displacement of SN-mDA neurons (at least not up to E18.5), suggesting that other Reelin sources could be important for mDA migration (*Prakash et al., 2009*). *Reelin* mRNA is expressed anterior to the SN, in the hypothalamus and ventral thalamus (*Alcántara et al., 1998*; *Allen Institute, 2015*). Moreover, it has been proposed that Reelin is transported from the striatum to the SN via axons in the striatonigral pathway (*Nishikawa et al., 2003*). Indeed, Reelin is expressed in the early differentiating cells in the striatum, but the striatonigral pathway is only established (E17 in rat) after the critical time period for SN-mDA migration (*Fishell and van der Kooy, 1987*; *Alcántara et al., 1998*).

## Reelin promotes the proportion of mDA neurons undergoing fast, directed migration

The visualization and tracking of a large population of migrating mDA neurons, and the subsequent categorization of the instantaneous soma speed of individual mDA neurons into slow, moderate and fast phases revealed that irrespective of their max-speed, mDA neurons spent a majority of their time at rest. During their migratory phase, mDA neurons move mostly at slow speed. Moderate-to-fast laterally-directed migration spurts that result in large displacements are infrequent and occur in only a subset of labeled mDA neurons during the time-window of imaging. Thus, mDA neurons migrate in two modes: in a frequent, slow mode and in infrequent, fast movements with a strong lateral orientation. A similar pattern of migration with variable instantaneous speeds and periods of rest has also been reported for newly generated granule cells in the dentate gyrus and for cortical projection neurons (*Simó et al., 2010*; *Wang et al., 2018*).

Comparing mDA tangential migration in the presence and absence of Reelin signaling, we observed that the duration of the individual migratory phases as well as average speed distribution of mDA neurons was comparable between control and *Dab1⁻/⁻* slices, while the likelihood of moderate-to-fast migration events was decreased in mDA neurons in *Dab1⁻/⁻* slices. In addition, a higher proportion of mDA neurons spent the entire imaging period at rest. Hence, Reelin promotes the likelihood with which moderate-to-fast migration spurts occur and increases the proportion of cells that enter a migratory phase.

Interestingly, the increased presence of activated DAB1 in cortical projection neurons – as a consequence of reduced ubiquitination and degradation in absence of the E3 ubiquitin Ligase Cullin-5 – leads to the opposite effect in the migratory behavior of these neurons: periods of rest are decreased and average as well as instantaneous speed is increased at late stages of cortical migration (E16.5) (*Simó et al., 2010*). This would be consistent with the role of Reelin that we observe in the migration of mDA neurons. In contrast, average speed appears not to be altered in cortical neurons of *reeler* mutants at this stage of development (*Chai et al., 2016*). Observation of cortical projection neurons in their multipolar-to-bipolar transition phase at E15.5 suggests yet another effect of Reelin: at this stage cortical neurons were observed to migrate faster in the absence of Reelin signaling while addition of exogenous Reelin slowed down migrating neurons, but only within the subventricular zone (*Britto et al., 2011*; *Britto et al., 2014*). Thus, even in the same neuronal population, Reelin signaling might have diverse effects on the speed of neuronal migration at different stages of migration.

## Reelin promotes a preference for directed migration

While moderate-to-fast migratory events are less likely in the *Dab1⁻/⁻* mDA population, individual moderate-to-fast *Dab1⁻/⁻* mDA neurons are equally laterally-directed as control mDA neurons. In contrast, 'slow' cells, which are weakly anisotropic in controls are significantly more isotropic in *Dab1⁻/⁻* slices. Whether this increase in isotropy in 'slow' cells reflects a general change in the behavior of this group or is specifically due to the population of 'fast' and 'moderate' cells that are now shifted into the 'slow' cell category cannot be resolved from our data. The loss of the laterally-directed slow movements might interfere with mDA neuron's ability to initiate moderate-to-fast, laterally-directed spurts. Indeed, mDA neurons have an aberrant orientation in E13.5 *reeler* brains

(*Bodea et al., 2014*). In the cortex, Reelin regulates orientation and cell polarity of multipolar neurons in the intermediate zone facilitating their switch to bipolar, glia-dependent migration (*Jossin and Cooper, 2011*; *Gärtner et al., 2012*; *Gil-Sanz et al., 2013*). Cortical projection neurons in their early phase of migration have been shown to deviate from radial migratory trajectories, in the absence of Reelin signaling as well as in the presence of exogenous Reelin (*Britto et al., 2011*; *Britto et al., 2014*; *Chai et al., 2016*). Reelin also promotes directionality during the radial migration of dentate gyrus cells (*Wang et al., 2018*). Interestingly, a recent study provides evidence that mDA neurons derived from induced pluripotent stem cells homozygous or heterozygous for a *REELIN* deletion show a disruption in their directed migratory behavior in neurosphere assays. Since the disruption occurs in absence of any organized tissue structure, Reelin signaling seems to modulate the ability of mDA neurons for directed migration independently of a specific pattern of Reelin protein deposition in the surrounding tissue (*Arioka et al., 2018*). In conclusion, Reelin appears to be a crucial factor in enabling SN-mDA neurons to initiate directed migration rather than a factor that guides SN-mDA neurons in a particular direction.

## Reelin signaling promotes stable morphologies in SN-mDA neurons

We show that moderate and fast movements of mDA neurons are strongly associated with bipolar morphologies both in control and *Dab1*$^{-/-}$ slices. Bipolarity is still predominant in slow phases, but about a third of the slow phases are associated with a multipolar morphology. In control slices, more than half of mDA neurons maintain a bipolar morphology throughout the imaging period, while about 40% transition between multipolar and bipolar morphologies. Only a small subset of cells (about 10%) stays multipolar at all timepoints. In absence of Reelin signaling, the percentage of transitionary cells is significantly increased, and the proportion of stable bipolar cells is decreased. Interestingly, the increase in the proportion of transitionary cells in *Dab1*$^{-/-}$ slices is particularly pronounced in the cell population that does not reach moderate-to-fast migration speeds and that is significantly more isotropic (data not shown) suggesting a correlation between loss of anisotropy in these cells and increased transitioning between bipolar and multipolar morphology. In transitionary cells of *Dab1*$^{-/-}$ slices, there is a significant increase in branch transitions at the soma and LP, a sign of decreased branch stability. Moreover, the length of the LP is significantly more variable in *Dab1*$^{-/-}$ than in control neurons. Thus, Reelin signaling appears to promote stability of morphologies once they have been adopted at specific phases of migration in mDA neurons.

In cortical neurons, Reelin appears to have multiple effects on cell morphology. In dissociated cortical neuronal cultures, Reelin signaling results in an increase in filopodia formation, likely via activation of Cdc42 (*Leemhuis et al., 2010*). Moreover, in presence of exogenous Reelin in organotypic slice cultures, projection neurons in the ventricular zone display a greater proportion of multipolar morphology, a phenotype concomitant with reduced migratory speeds (see above, *Britto et al., 2014*). In contrast, LP morphology of migrating cortical neurons is comparable in presence and absence of Reelin signaling when these neurons first contact the marginal zone of the cortex, but Reelin signaling is required to maintain this morphology and a stable LP during the final somal translocation step of these neurons (*Franco et al., 2011*; *Chai et al., 2016*). Finally, a recent study showing the phosphorylation of DAB1 via the Netrin receptor DCC has reported an increase in multipolar neurons in the subventricular zone of *Dcc* knockout cortex (*Zhang et al., 2018*). In summary, depending on location, concentration, and sub-cellular localization, Reelin and DAB1 can have differing effects on the morphology of migrating neurons.

An indirect regulation of morphology by Reelin signaling has been reported in tangentially-migrating cortical interneurons. In interneurons, branching of LPs aids in precise sensing of the extracellular environment during chemotaxis (*Martini et al., 2009*). In the inverted *reeler* cortex, interneurons display a significantly higher number of branch nodes and higher length of LPs than interneurons in control brains (*Yabut et al., 2007*). This aberrant morphology is accompanied by their ectopic location in cortical layers. Since interneurons do not directly require Reelin signaling for their migration, it is likely that their aberrant morphology in the *reeler* cortex are an indirect effect of their altered position. As we observe similar effects on cell morphology in *Dab1*$^{-/-}$ mDA neurons, the aberrant mDA neuronal morphology may be a consequence of an increased necessity to scan the environment for guidance cues in ectopic medial positions rather than a direct downstream effect of Reelin.

## Reelin downstream signaling in SN-mDA neurons

It has previously been demonstrated that the regulation of CDH2 via the Crk/CrkL-C3G-Rap1 pathway at the cell surface is important for the effect of Reelin on the polarity of cortical projection neurons during their migration (*Franco et al., 2011*; *Sekine et al., 2012*; *Park and Curran, 2008*; *Voss et al., 2008*). Cofilin1 has been shown to stabilize the LPs of migrating cortical neurons downstream of Reelin signaling-activated LIMK1 (*Chai et al., 2009*; *Chai and Frotscher, 2016*). In addition to its effect on these actin cytoskeleton modulators, Reelin has also been described to alter microtubule stability in the developing cortex (*Meseke et al., 2013*). However, we demonstrate here that expression and/or phosphorylation levels of these Reelin downstream effectors are not obviously altered in mDA neurons in the absence of Reelin signaling. Moreover, we could not detect differences in the expression pattern of EB3 and acetylated α-tubulin, which are indicators for microtubule dynamics and stability, respectively (*van de Willige et al., 2016*; *Fernández-Barrera and Alonso, 2018*). We can however not exclude that the immunoblotting of ventral midbrain tissue and immunostaining on sections may miss subtle defects in the regulation of actin- or microtubule dynamics that underlie the migratory abnormalities in absence of *Dab1*.

Other signaling events that influence cortical migration downstream or in parallel to Reelin signaling are mediated through integrin α5ß1 or the Netrin1-DCC pathway. The knockdown of integrin α5ß1 in cortical neurons affects apical process stability during terminal translocation suggesting that additional adhesion molecules may be recruited by Reelin signaling (*Sekine et al., 2012*). In the cortex, both CDH2 and integrin α5ß1 act downstream of Reelin, with integrin α5ß1 anchoring the leading tip of terminally translocating neurons in the marginal zone and CDH2 regulating the subsequent cell movements (*Sekine et al., 2014*). Interestingly, integrin α5ß1 has been shown to be important for stabilizing neurite extensions of mDA neurons in vitro. Whether it plays a general role in stabilizing neuronal processes in mDA neurons, including LPs, and in mDA migration has not been explored (*Izumi et al., 2017*). As mentioned above, cross talk between Netrin1-DCC and Reelin–Dab1 pathways has been reported in migration of cortical projection neurons (*Zhang et al., 2018*). The Netrin1–DCC pathway is also important for proper localization of SN-mDA neurons during development (*Xu et al., 2010*; *Li et al., 2014*). Though the effect on mDA distribution induced by *Dcc* inactivation differs from the effect caused by *Dab1* inactivation, it is still possible that effectors downstream of the Netrin1-DCC pathway, such as focal adhesion kinase my play a role in mediating Reelin signal in mDA neurons (*Zhang et al., 2018*).

## Conclusion

Here we provide a detailed characterization of the migratory modes and cellular morphologies underlying SN-mDA tangential migration to gain a detailed understanding of SN formation and to open the door to further investigations of the molecular mechanisms of mDA migration. Moreover, we demonstrate that Reelin directly regulates lateral, tangential migration of mDA neurons by promoting the lateral directionality of small, slow movements, increasing the frequency of laterally-directed moderate-to-fast migration events that cover larger distances and stabilizing morphology of mDA neurons. We thus provide new mechanistic insight into how Reelin signaling regulates the formation of the SN and how Reelin signaling controls tangential migration.

## Materials and methods

**Key resources table**

| Reagent type (species) or resource | Designation | Source or reference | Identifiers | Additional information |
|---|---|---|---|---|
| Genetic reagent (M. musculus) | Shh*CreER* | PMID: 15315763 | RRID: MGI:J:92504 | Clifford Tabin, Harvard University |
| Genetic reagent (M. musculus) | Slc6a3*Cre* | PMID: 17227870 | RRID: MGI:3702746 | Nils-Görran Larsson, Max Planck Institute for Biology of Aging, Cologne, Germany |

*Continued on next page*

*Continued*

| Reagent type (species) or resource | Designation | Source or reference | Identifiers | Additional information |
|---|---|---|---|---|
| Genetic reagent (M. musculus) | *Dab1$^{flox}$* | PMID: 21315259 | RRID: MGI:5141401 | generated by Ulrich Müller, Johns Hopkins University, Baltimore, USA; obtained from Amparo Acker-Palmer, University of Frankfurt |
| Genetic reagent (M. musculus) | *Dab1$^{-/-}$* | PMID: 21315259, recombined flox allele | RRID: MGI:5141401, recombined flox allele | generated by Ulrich Müller, Johns Hopkins University, Baltimore, USA; obtained from Amparo Acker-Palmer, University of Frankfurt |
| Genetic reagent (M. musculus) | *ROSA$^{loxP-STOP-loxP-EYFP}$* | PMID: 11299042 | RRID: MGI:J:80963 | Frank Constantini, Columbia University |
| Antibody | sheep anti-DIG-AP Fab fragments | Roche | RRID:AB_514497 | ISH: 1:5000 |
| Antibody | Goat anti-OTX2 | Neuromics | RRID:AB_2157174 | IHC: 1:5000 |
| Antibody | goat anti-Reelin | R and D systems | RRID:AB_2253745 | IHC: 1:50 |
| Antibody | mouse anti-α-Tubulin | Merck | RRID:AB_477579 | IHC: 1:500 |
| Antibody | mouse anti-α-Tubulin, acetylated | Merck | RRID:AB_477585 | IHC: 1:500 |
| Antibody | mouse anti-TH | Merck | RRID:AB_2201528 | IHC: 1:500 |
| Antibody | rabbit anti-Calbindin | Swant | RRID:AB_2314067 | IHC: 1:5000 |
| Antibody | rabbit anti-CDH2 | Abcam | RRID:AB_444317 | WB: 1:500 |
| Antibody | rabbit anti-DAB1 | Dr. Brian Howell | *Howell et al., 1997* | IHC: 1:5000 (used with TSA kit) |
| Antibody | rabbit anti-EB3 | Abcam | RRID:AB_880026 | IHC: 1:250 |
| Antibody | rabbit anti-GIRK2 | Alamone Labs | RRID:AB_2040115 | IHC: 1:400 |
| Antibody | rabbit anti-GFP | Thermo Fischer | RRID:AB_221569 | IHC:1:400 |
| Antibody | rabbit anti-SOX6 | Abcam | RRID:AB_1143033 | IHC: 1:500 |
| Antibody | rabbit anti-TH | Merck | RRID:AB_390204 | IHC: 1:500 |
| Antibody | rat anti-GFP | Nalacai | RRID:AB_10013361 | IHC: 1:1500 |
| Antibody | rabbit anti-Cofilin1 | Kindly provided by Prof. Dr. Walter Wittke | | WB: 1:5000 |
| Antibody | rabbit anti-p-Cofilin1 (ser3) (77G2) | Cell signaling | RRID:AB_2080597 | WB: 1:1000 |
| Antibody | rabbit anti-p-LIMK1 (Thr508)/LIMK2 (thr505) | Cell signaling | RRID:AB_2136943 | WB: 1:500 |
| Antibody | rabbit anti-LIMK1 | Cell signaling | RRID:AB_2281332 | WB: 1:100 |
| Antibody | anti rabbit-HRP-linked | Cell signaling | RRID:AB_2099233 | WB: 1:1000 |
| Antibody | donkey anti-rabbit Alexa 488 | Thermo Fischer | RRID:AB_2535792 | IHC: 1:500 |
| Antibody | donkey anti-rabbit Alexa 350 | Thermo Fischer | RRID:AB_2534015 | IHC: 1:500 |
| Antibody | donkey anti-mouse Alexa 488 | Thermo Fischer | RRID:AB_141607 | IHC: 1:500 |
| Antibody | donkey anti-rat Alexa 488 | Thermo Fischer | RRID:AB_2535794 | IHC: 1:500 |
| Antibody | donkey anti-goat Alexa 488 | Jackson ImmunoResearch | RRID:AB_2336933 | IHC: 1:500 |

*Continued*

| Reagent type (species) or resource | Designation | Source or reference | Identifiers | Additional information |
|---|---|---|---|---|
| Antibody | donkey anti-rabbit Cy3 | Jackson ImmunoResearch | RRID:AB_2307443 | IHC: 1:200 |
| Antibody | donkey anti-mouse Cy3 | Jackson ImmunoResearch | RRID:AB_2340813 | IHC: 1:200 |
| Antibody | donkey anti-goat Cy3 | Jackson ImmunoResearch | RRID:AB_2307351 | IHC: 1:200 |
| Antibody | donkey anti-rabbit Biotin | Jackson ImmunoResearch | RRID:AB_2340593 | IHC: 1:200 |
| Recombinant DNA reagent | in situ mRNA probe: *Reln* | pCRII-Topo vector, 0.64 kb(cDNA) RELN inserted fragment | | Joachim Herz, UT Southwestern, Dallas, USA |
| Recombinant DNA reagent | in situ mRNA probe: *Lmo3* | pCMV-SPORT6 vector, 3 kb(partial cDNA) inserted fragment | Image clone: 4913098; accession #BC034128 | Source Bioscience, Berlin, DE |
| Commercial assay or kit | TSA Plus fluorescence kit | Perkin Elmer | NEL744001KT | |
| Chemical compound, drug | Tamoxifen | Merck | T5648 | |
| Software, algorithm | Quantity One | BioRAD | | Gel documentation |
| Software, algorithm | Zen Blue 2012 | Carl Zeiss | | Image acquisition |
| Software, algorithm | Zen Black 2012 | Carl Zeiss | | Image acquisition |
| Software, algorithm | Leica Application Suite X 4.13 | Leica microsystems | | Image acquisition |
| Software, algorithm | Fiji/ImageJ 1.51 n | Wayne Rasband. National Institutes of Health | | Image processing |
| Software, algorithm | Adobe Photoshop CS3 | Adobe Systems | | Image processing |
| Software, algorithm | Imaris 8.3.1 | Bitplane | | Image processing |
| Software, algorithm | MatLab R2017b | MathWorks | | Image processing and data analysis |
| Software, algorithm | Affinity Designer 1.5.5 | Serif | | Image editing |
| Software, algorithm | Image Lab 6.0 | BioRAD | | Immunoblot quantification and analysis |
| Software, algorithm | GraphPad Prism 7.0 | GraphPad Software | | Statistical analysis |

## Mouse lines

*Dab1^flox^* and *Dab1^del^* mice (*Franco et al., 2011*) were kindly provided by Dr. Ulrich Müller, Johns Hopkins University, Baltimore and Dr. Amparo Acker-Palmer, University of Frankfurt. *Dab1* CKO mice (genotype: *Scl6a3^Cre/+^, Dab1^flox/del^*) were generated by crossing *Dab1^flox/flox^* mice with *Scl6a3-Cre/+, Dab1^+/del^* mice (*Ekstrand et al., 2007*). *Dab1^del/+^* mice were used to generate complete knockouts of Dab1 (*Dab1^-/-^*). *Scl6a3^Cre/+^* mice were crossed with *ROSA^loxP-STOP-loxP-EYFP^* mice (*Srinivas et al., 2001*) to analyze the timing and extent of recombination. Mosaic labelling of migrating mDA neurons was achieved by crossing *Shh^CreER^* mice (*Harfe et al., 2004*) with *ROSA^loxP-STOP-loxP-EYFP^* mice. Day of vaginal plug was recorded as E0.5. Mice were housed in a controlled environment, with 12 hr light/night cycles and *ad libidum* availability of food and water.

All experiments were performed in strict accordance with the regulations for the welfare of animals issued by the Federal Government of Germany, European Union legislation and the regulations of the University of Bonn. The protocol was approved by the Landesamt für Natur, Umwelt und Verbraucherschutz Nordrhein-Westfalen (Permit Number: 84-02.04.2014.A019).

## Tamoxifen

Tamoxifen (75 mg/kg body weight) was administered by gavage to pregnant dams at E8.5 to label SN-mDA neurons (*Bodea et al., 2014*). TM (Sigma Aldrich) was prepared as a 20 mg/mL solution in corn oil (Sigma Aldrich), with addition of progesterone (Sigma Aldrich, 5 mg/mL) to reduce miscarriages.

## Immunohistochemistry

Pregnant dams were sacrificed by cervical dislocation. Embryos were dissected in ice cold PBS. Heads (E13.5 – E15.5) or brains (E16.5 – E18.5) were fixed in 4% paraformaldehyde (PFA) for 2–3 hr at room temperature (RT). Adult mice were anesthetized with isofluorane, perfused transcardially with phosphate buffered saline (PBS), followed by 4% PFA. Tissue was cryopreserved in OCT Tissue Tek (Sakura), embryonic tissue was cryosectioned at 14 μm, adult brains were cryosectioned at 40 μm thickness. Immunostaining was essentially performed as previously described (*Blaess et al., 2011*).

For immunostainings, sections were fixed briefly in 4%PFA (5 min at RT), followed by 1 hr incubation in 10% NDS in 0.1% Triton in PBS (0.1% PBT). Sections were incubated with primary antibody for 4 hr at RT (anti-α-Tubulin and Mouse anti-acetylated α-Tubulin) or overnight at 4°C (all other primary antibodies) in 3% NDS in 0.1% PBT. Sections were washed 3X in 0.1%-PBT and incubated for 2 hr in secondary antibody in 3% NDS in 0.1% PBT before mounting with Aqua Polymount (Polysciences Inc.).

For the detection of SOX6, antigen retrieval was carried out in 0.1M EDTA for 30 min at 65°C before blocking, and Cy3-Streptavidin amplification was used with biotinylated donkey anti-rabbit antibody. To improve detection of DAB1 with rabbit anti-DAB1 antibody in E15.5 embryonic sections, a tyramide signal amplification (TSA) was carried out with the TSA kit (Perkin Elmer) as follows: Sections were blocked in the TSA kit blocking solution for 1 hr followed by incubation with rabbit anti-DAB1 antibody (1:5000, (*Howell et al., 1997*) in 0.1% TBST (Tris buffered saline with 0.1% Triton) overnight at 4°C. After a washing step in TBST, sections were incubated for 2 hr at RT with biotinylated donkey anti-rabbit in TBST, followed by another washing step and incubation with HRP conjugated Streptavidin (1: 1000) in TBST for 1 hr at RT. Sections were again washed with TBST and incubated for 10 min with TSA detection reagent. After additional washing steps in TBST and 0.1% PBT sections were co-stained for TH following the standard immunostaining protocol. A complete list of primary and secondary antibodies is presented in Key Resources Table.

## Immuno blotting

Ventral midbrain of control and *Dab1*$^{-/-}$ embryos were were isolated at E14.5 and snap-frozen in liquid nitrogen. Tissue extraction was performed with RIPA buffer (Sigma, R0278) supplemented with 1x Halt protease and phosphatase inhibitor (Thermofischer Scientific, 78442) on ice according to the manufacturer's instructions. Protein concentrations were determined by BCA assay (Thermofischer Scientific) using a BSA calibration curve. Protein supernatant was mixed with 4x LDS buffer and loaded on a 4–12% Bis Tris gel (NuPAGE, NP0335BOX). Protein was blotted on a PVDF membrane, blocked for 1 hr at RT and incubated with primary antibody overnight. After washing with TBST, membrane was incubated with a corresponding horse radish peroxidase (HRP) coupled secondary antibody. Membrane was washed with TBST and visualization of immunoreactive proteins was conducted with a chemiluminescent HRP substrate solution (Super signal femto, Thermofischer Scientific/Western HRP substrate, Merck Millipore) using a chemiluminescent imager (Chemidoc, Bio-Rad). Bound proteins were removed using 1x Western blot stripping buffer (2% SDS, 60,02 mM Tris (pH 6.8), 100 mM ß-mercaptoethanol) and immunodetection was repeated. For quantification, densitometric analysis was performed, normalization was carried out with total protein as loading control (Amido Black, Sigma Aldrich) using the software Image Lab (Bio-Rad).

## In situ hybridization

Sections were post-fixed in 4% PFA for 10 min, rinsed in PBS and acetylated in 0.1 M TEA (triethanolamine)-HCl with 125 µL acetic anhydride for 5 min with stirring. Sections were washed in PBS and briefly dehydrated in 70%, 95% and 100% ethanol (EtOH). 1 µg of RNA probe was added to 1 mL hybridization buffer and incubated for 2 min at 80°C. Sections were air-dried and transferred to a humidified hybridization cassette. A 1:1 mixture of formamide and $H_2O$ was used as humidifying solution. 300 µL hybridization solution containing RNA probe was added to each slide, slides were covered with RNase-free coverslips and incubated at 55°C overnight. On the following day, coverslips were removed in prewarmed 5X SSC. To reduce unspecific hybridization, sections were incubated in a 1:1 solution of formamide and 2X SSC (high stringency wash solution) for 30 min at 65°C. Sections were then washed with RNAse buffer, containing 0.1% RNase A at 37°C for 10 min to remove non-hybridized RNA. Sections were washed twice with high stringency solution for 20 min at 65°C, once with 2X SSC and once with 0.1X SSC for 15 min at 37°C. Sections were placed in a humidified chamber and incubated with 10% normal goat serum in 0.1% PBS-Tween (blocking solution) for 1 hr at RT. Sections were incubated with anti-DIG-AP Fab fragments (diluted 1:5000 in 1% goat serum in 0.1% PBS-Tween) for 3 hr at RT, or overnight at 4°C. Sections were washed several times with 0.1% PBS-Tween, followed by two washes in NTMT buffer (containing 1 mg/mL levamisole to reduce background of endogenous alkaline phosphatase activity) for 10 min at RT. Sections were incubated in BM purple, a substrate for alkaline phosphatase (with 0.5 mg/mL levamisole) at RT until signal was observed. The chromogenic reaction was stopped by a 10 min incubation in TE buffer at RT. Sections were then washed in PBS, and immunostained for TH.

## Image acquisition of fixed cryosections

Embryonic and adult sections were imaged at an inverted Zeiss AxioObserver Z1 microscope equipped with an ApoTome. Fluorescence images were acquired with Zeiss AxioCam MRm 1388 × 1040 pixels (Carl Zeiss). At 10X (EC PlnN 10x/0.3, Carl Zeiss) and 20X (EC PlnN 20x/0.5, Carl Zeiss) magnifications, tile images were acquired with conventional epifluorescence. ApoTome function was used to acquire tile images and z-stacks at 40X (Pln Apo 40x/1.3 Oil, Carl Zeiss) and 63X (Pln Apo 63x/1.4 Oil, Carl Zeiss) magnifications. In situ hybridized sections were imaged with transillumination (AxioCam MRc, 1300 × 1030 pixels, Carl Zeiss) at the AxioObserver Z1 setup. Images were stitched with Zen blue software (Zeiss, 2012). Sections stained with Alexa 649 secondary antibody, and 63X confocal images were imaged at a Leica SP8 confocal microscope and stitched with Leica PC suite (Leica, 2014)

## Organotypic slice culture and time lapse imaging

Organotypic slice cultures were generated as previously described (*Bodea and Blaess, 2012*). Slices were placed on Millicell membrane inserts (Merck) and incubated for 6–12 hr at 37°C, 5% $CO_2$, before imaging. Slices were briefly examined at a Zeiss Axioobserver microscope with conventional epifluorescence. Healthy slices, with well defined, strongly fluorescent cells, were chosen for two-photon excitation imaging. Slices on their membrane inserts were transferred to µ-Dish imaging dishes (Ibidi) containing 750 µL of prewarmed, fresh culture medium (5 mL Hank's balanced salt solution, 9 mL DMEM high glucose (Sigma Aldrich), 5 mL horse serum, 200 µL Penicillin/Streptomycin for 20 mL of culture medium). Slices were imaged at 32X magnification (C-Achroplan 32x/0.85, Carl Zeiss) with an inverted, two-photon Zeiss LSM 710 NLO microscope, equipped with temperature and $CO_2$ control (Pecon). The microscope setup and the 32X water immersion objective were pre-heated for 8 hr before time lapse experiments. Images were acquired using 920 nm for excitation with a laser power of 5–10% (Laser: Chameleon UltraII, Coherent). A total of 3 control ($Shh^{CreER/+}$, $Rosa26^{lox-stop-lox\ YFP/+}$, $Dab^{+/+}$ or $Shh^{CreER/+}$, $Rosa26^{lox-stop-lox\ YFP/+}$, $Dab1^{del/+}$) and $Dab1^{-/-}$ slices ($Shh^{CreER/+}$, $Rosa26^{lox-stop-lox\ YFP/+}$, $Dab1^{del/del}$), across four litters, were imaged as described. Of the six slices analyzed, 3 control and 2 $Dab1^{-/-}$ were imaged for 4.3 hr while one $Dab1^{-/-}$ slice was imaged for 2.6 hr.

All imaged slices were post-stained with TH to confirm that the region imaged was within the dopaminergic domain (*Bodea et al., 2014*). Organotypic slice cultures were fixed in 4% PFA for 1 hr at RT, then rinsed in PBS and 0.3% PBT for 10 min. Slices were incubated in blocking solution (10% NDS in 0.3% PBT) at RT for 2 hr, or overnight at 4°C. After blocking, slices were incubated with

primary antibody solution (3% NDS in 0.3% PBT) for 24–48 hr at 4°C. The following primary antibodies and dilutions were used: rabbit anti-TH (1:500), rat anti-GFP (1:1000). Slices were washed in 0.3% PBT and then incubated in secondary antibody solution (3% NDS in 0.3% PBT), at RT for 4 hr, or overnight at 4°C. Secondary antibodies donkey anti-rabbit Cy3 (1:200) and donkey anti-rat Alexa 488 (1:500) were used. All steps were carried out in a 6-well plate.

### Immunostaining and clearing of whole mount embryonic brains

Brains from E13.5 embryos were fixed in 4% PFA for 4 hr at room temperature, or overnight at 4°C. Brains were washed with PBS, 0.3% PBT, and incubated with blocking solution (10%NDS in 0.3% PBT) overnight at 4°C. The brains were incubated with primary antibodies: rabbit anti-TH (1:500) and rat anti-GFP (1:1000) at 4°C for 2 days. Next, the primary antibody solution was removed and the brains were washed three times with 0.3% PBT at RT for 15 min. The tissue was incubated with secondary antibodies: donkey anti-rat IgG-DyLight 647 (1:100) and donkey anti-rabbit Cy3 (1:200) at RT for 1 day. Subsequently, the tissue was washed three times with 0.3% PBT and three times with PBS for 20 min. All washing steps and antibody solutions preparation were performed using 0.3% PBT. All steps were carried out in 24-well plates.

Tissue clearing was carried out as described previously (*Schwarz et al., 2015*). The procedure was modified for embryonic tissue as described here: After immunostaining, brains were incubated in increasing concentrations (30%, 50%, 70%) of tert-butanol (pH 9.5) for 4 hr at RT followed by 96% and 100% tert-butanol (pH 9.5) for 4 hr at 33°C. Brains were then incubated overnight in a triethylamine pH-adjusted 1:1 mixture of benzyl alcohol/benzyl benzoate (BABB, pH 9.5) at 33°C. Brains were stored in clearing solution at 4°C and imaged within 1 week of clearing. Whole mount brains were imaged in clearing solution with a 20X BABB dipping objective (Olympus) at a Leica SP8 upright microscope. Multi-channel image acquisition of the whole brain (4–6 tiles, 150–200 z-steps, step-size = 1.5 µm) took 30–70 hr and resulted in multichannel datasets of large sizes (20–80 GB). Voxel size of thus acquired images was 0.432 µm X 0.432 µm X 1.5 µm. Individual tiles at each z-step were stitched together using the Leica SP8 PC suite (Leica, 2014).

### Analysis of cell distribution in fixed sections

Mediolateral distribution of mDA neurons was analyzed for n ≥ 3 animals at each time point of analysis (E15.5, E18.5 and P21-30) by constructing normalized bins spanning the entire TH-positive domain. Since we observed that in both, *Dab1* CKO and *Dab1* $^{-/-}$ mice, a few TH-positive cells of the lateral most SN lateralis were consistently present (yellow arrowheads *Figure 1D,E,H,I*), we defined the mediolateral bins by quadrisecting a line extending from the midline to the lateral most TH positive cells (*Figure 4B*). The fraction of the total number of TH positive cells in each mediolateral bin was evaluated for control, *Dab1* CKO and *Dab1*$^{-/-}$ brains.

The number of mDA neurons in SN or VTA was determined by defining the anatomical area of the SN and VTA at three rostrocaudal levels (*Franklin and Paxinos, 2013*). The number of mDA neurons in these two regions were counted and the numbers were normalized for the total number of mDA neurons (SN +VTA) (*Figure 1—figure supplement 4*).

### Analysis of TH$^+$ projections in fixed sections

10X epifluorescence images of 40 µm thick free-floating slices stained for TH were acquired at an inverted Z1 Axioobserver microscope. Raw integrated intensity of the TH$^+$ striatal projections (dorsal and ventral) was calculated at three rostrocaudal levels of the striatum. The raw integrated intensity was normalized for the background (intensity of non-TH$^+$ areas) and for the area (of TH$^+$ projections) (*Figure 3—figure supplement 1*).

### Speed and trajectories of migrating mDA neurons

To prevent any bias in selection of cells for tracking, and to track a large number of neurons in 3D in our two-photon time lapse datasets, we used the semi-automatic plugin TrackMate in Fiji (*Tinevez et al., 2017*). Before soma detection, a 3 × 3 median filter was applied by the TrackMate plugin, to reduce salt and pepper background noise. Soma detection was carried out using the Laplacian of Gaussian (LoG) detector in TrackMate. The soma detected by the TrackMate plugin were automatically linked across time, in 3D, by using the linear assignment problem (LAP) tracker in

TrackMate (*Tinevez et al., 2017*; *Jaqaman et al., 2008*). After automatic tracking, the track scheme view in TrackMate was used to check the accuracy of each track by eye. Spurious tracks were deleted and missed detections were added using the manual tracking mode in TrackMate. Excel files from the TrackMate plugin were imported into MatLab. 3D soma velocity was obtained at every time point (in units of µm/hr) of the analysis (starting t = 10 min) as the change in soma position vector between the previous frame and the current frame, divided by the time duration (0.167 hr) between frames. This data was used to generate probability histograms for average soma speed, maximum soma speed, time spent at rest (defined as soma speed <10 µm/ hr), time spent in slow migration (soma speed between 10 and 30 µm/ hr), time spent in medium-fast migration (30–60 µm/ hr) and time spent in fast migration (soma speed >60 µm/ hr). Categories for rest, slow, medium-fast and fast speeds were defined for the purpose of easy visualization of data and were based on 25% percentile (10 µm/hr) and 75% percentile speeds (30 µm/hr) of $Dab1^{-/-}$ population.

Cell trajectory angles were measured in 2D as the angle between midline (positive y-axis in the image) and the line joining the first and final soma positions. Cells that moved with maximum speeds of less than 10 µm/hr were excluded from the trajectory analysis as they were categorized as being at rest. Statistics on trajectory angles were performed with CircStat: a MatLab toolbox (*Berens, 2009*).

Only cells for which the soma were detected at all time points of imaging were included in the analysis. Using this approach, we tracked 806 cells in slices from control mice ($Shh^{CreER/+}$, $Rosa26^{lox-stop-lox\ YFP/+}$, $Dab^{+/+}$ or $Shh^{CreER/+}$, $Rosa26^{lox-stop-lox\ YFP/+}$, $Dab1^{-/+}$) and 844 cells from $Dab1^{-/-}$ mice ($Shh^{CreER/+}$, $Rosa26^{lox-stop-lox\ YFP/+}$, $Dab1^{-/-}$), across three slices and acquired their speed and trajectory profiles. Each cell (and track) had a unique ID assigned by the TrackMate plugin. These cell IDs were used to identify and locate individual cells in the slice for further analysis.

## Morphology analysis of migrating mDA neurons

We restricted our morphological analysis to n = 150 control (70 fast, 40 medium-fast and 40 slow cells), and 129 $Dab1^{-/-}$ (49 fast, 40 medium-fast and 40 slow) cells. We observed that $Dab1^{-/-}$ cells continuously extended protrusions in slices and this made it difficult to unambiguously assign processes to individual cells as imaging progressed. Hence, we examined the morphology of each cell, in 3D, for the first 18 frames of imaging. Cell soma was defined as the spot detected/assigned to the cells in the TrackMate plugin. Analysis was done manually, by rendering individual neurons in 4D (3D projection over all time frames) in ImageJ and recording the number of primary processes (arising from the soma) and secondary processes at each time point. A cell was defined as bipolar when fewer than two processes were observed arising directly from the soma. The appearance/disappearance of any branch was regarded as a branch transition. At each time point, the morphology of the cell, and the number of branch transitions, was manually annotated to the spot position data of the cell in excel sheets exported from TrackMate. In addition, 20 control and $Dab1^{-/-}$ cells were randomly chosen for tracing in 3D. These neurons were traced manually in simple neurite tracer (SNT) plugin of Fiji. Tracings were carried out, at each time point individually, for the first 18 frames of imaging. Fills of traced neurons were generated semi-automatically in the SNT plugin. Fill thickness was decided by eye but was maintained across all time points for a cell. Maximum intensity projections were also generated for the 3D segmentation fills. SNT traces were also used to measure length of the leading process in 3D.

## Statistical analysis

Statistical significance of mediolateral distributions of TH$^+$ mDA neurons in control, $Dab1$ CKO and $Dab1^{-/-}$ adult and embryonic brains were assessed by two-way ANOVA with Tukey's correction for multiple comparisons (n = 6 animals/genotype, at P30 and n = 4 animals/genotype at E18.5). At E15.5, mediolateral distribution of TH$^+$ mDA neurons and P30 TH$^+$ GIRK2$^+$ mediolateral distributions in control and $Dab1$ CKO brains were assessed for statistical significance by Student's t-test. All non-parametric distributions were analyzed with Mann-Whitney's non-parametric rank test or Kalmogrov-Smirnov test (mentioned in figure legends) in Prism 7/MatLab. Circular variables were analyzed with the CircStat toolbox for MatLab (*Berens, 2009*). Angle distribution in populations were compared using Kuiper's test for circular variables (*Berens, 2009*).

## Acknowledgements

This work was supported by the Maria von Linden-Program and BONFOR (both University of Bonn, to SB), the German Research Foundation (BL 767/2–1, BL 767/3–1, BL 767/4–1, SFB 1089 to SB) and a German Academic Exchange Service doctoral fellowship (to ARV). We thank Brian Howell for providing the DAB1 antibody; Walter Witke for providing the Cofilin1 antibody, Joachim Herz for providing the *Reelin* in situ probe; Ulrich Müller and Amparo Acker-Palmer for providing the $Dab1^{del}$ and $Dab1^{flox}$ mouse lines; Nils-Göran Larsson for providing the $Scl6a3^{Cre}$ mouse line, Donato Di Monte and Michael Helwig for assistance with two-photon imaging; the UKB microscopy core facility for support with imaging, Jonas Doerr, Martin Schwarz and Anke Leinhaas for initial support with clearing and imaging of whole-mount brains; Petra Mocellin, Kilian Berendes and Philipp Grunwald for technical support and Gabriela Bodea, David Greenberg and Marianna Tolve for critical reading of the manuscript.

## Additional information

### Funding

| Funder | Grant reference number | Author |
| --- | --- | --- |
| Deutscher Akademischer Aus-tauschdienst | Graduate Student Fellowship | Ankita Ravi Vaswani |
| Rheinische Friedrich-Wilhelms-Universität Bonn | Gradute Student Support (BONFOR) and Fellowship for Mid-Career Scientists (Maria von Linden-Programm) | Ankita Ravi Vaswani Sandra Blaess |
| Seventh Framework Pro-gramme | FP7-HEALTH-F4-2013- 738 602278-Neurostemcellrepair | Beatrice Weykopf Oliver Brüstle |
| Deutsche Forschungsge-meinschaft | BL 767/2-1 | Sandra Blaess |
| Deutsche Forschungsge-meinschaft | BL 767/4-1 | Sandra Blaess |
| Deutsche Forschungsge-meinschaft | BL 767/3-1 | Sandra Blaess |
| Deutsche Forschungsge-meinschaft | SFB 1089 | Sandra Blaess |

The funders had no role in study design, data collection and interpretation, or the decision to submit the work for publication.

### Author contributions

Ankita Ravi Vaswani, Conceptualization, Data curation, Formal analysis, Validation, Investigation, Visualization, Methodology, Funding acquisition, Writing—original draft, Writing—review and editing; Beatrice Weykopf, Cathleen Hagemann, Investigation, Methodology, Writing—review and editing; Hans-Ulrich Fried, Resources, Methodology, Writing—review and editing; Oliver Brüstle, Resources, Funding acquisition; Sandra Blaess, Conceptualization, Resources, Data curation, Formal analysis, Supervision, Funding acquisition, Validation, Methodology, Project administration, Writing-draft, Writing—review and editing

### Author ORCIDs

Ankita Ravi Vaswani (iD) https://orcid.org/0000-0001-5015-8525
Sandra Blaess (iD) http://orcid.org/0000-0002-1898-4891

### Ethics

Animal experimentation: This study was performed in strict accordance with the regulations for the welfare of animals issued by the Federal Government of Germany, European Union legislation and

the regulations of the University of Bonn. The protocol was approved by the Landesamt für Natur, Umwelt und Verbraucherschutz Nordrhein-Westfalen (Permit Number: 84-02.04.2014.A019).

### Decision letter and Author response
Decision letter https://doi.org/10.7554/eLife.41623.sa1
Author response https://doi.org/10.7554/eLife.41623.sa2

## Additional files

### Supplementary files
- Source code 1. Tracking_data_analysis.
- Transparent reporting form

### Data availability
All data generated or analysed during this study are included in the manuscript and supporting files. Source data files have been provided for Figures 1,4,5,6,8,9,10.

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
