## [Decision Letter]

Thank you for submitting your article "Formation of the substantia nigra requires Reelin-mediated fast, laterally-directed migration of dopaminergic neurons" for consideration by eLife. Your article has been reviewed by three peer reviewers, and the evaluation has been overseen by a Reviewing Editor and Marianne Bronner as the Senior Editor. The following individual involved in review of your submission has agreed to reveal their identity: Christine Metin (Reviewer #1).

The reviewers have discussed the reviews with one another and the Reviewing Editor has drafted this decision to help you prepare a revised submission.

Summary:

In this study the authors examined SN formation in the midbrain, and specifically examined the lateral migration of mDA neurons away from the midline and the VTA. Using mutant mice, the same investigators and others previously showed that Reelin signaling is required for this particular event. Here using new *Dab1* conditional knock out mice in DA neurons they demonstrate that laterally migrating mDAs cell-autonomously require Reelin signaling; they then characterize in detail the modes of migration of these neurons and establish that *Dab1* is required to promote movement and especially fast, laterally-oriented migration. Finally, they show that bipolar morphology correlates with this fast mode of migration.

The paper clearly identifies and characterizes a cellular behavior responsible for the spatial segregation of two populations of DA neurons and the formation of two nuclei with distinct functions. Overall, the study is solid, employing advanced genetic and imaging techniques to carefully document the lateral migration of mDA neurons, and the specific requirement of *Dab1* (and by implication Reelin signaling) in this process. The rigor of this descriptive study will lay a strong foundation for future studies that address mechanism in greater detail.

Essential revisions:

The reviewers appreciated the careful descriptive nature of the study and agreed that substantially advancing the mechanism is likely beyond the scope of the current study. Nonetheless, there were two experiments that the reviewers felt would strengthen the study within its current bounds, and several points that could be addressed in the text. These are as follows:

1) About cellular mechanism: Wild-type and *Dab1^-/-^* SN-mDA neurons strongly differ by their morphology. Immunoblotting did not reveal any differences, but immunoblotting might miss subtle defects at the origin of those migratory abnormalities. In contrast, the organization of the microtubule cytoskeleton, especially the relative abundance of stable and dynamic microtubules could provide clues for understanding the migratory defects observed in *Dab1^-/-^* SN-mDA neurons. Such analysis can be performed by immunostaining and will give a precise information.

2) About the details of the DA populations involved in the phenotype: The authors mention that the phenotype is most severe at mid levels. In general, even at mid levels, the phenotype is subtle with only approximately 10-15% of the neurons being mislocalized (Figure 3C, bin 1 and 3). Is there even a phenotype in more rostral sections? The *Dab1* CKO should be analyzed/quantified along the rostrocaudal axis. If one counts total SN and VTA neurons (rather than mediolateral bins), what is the net reduction in SN, and what is the net gain in VTA DA neurons? A related question is why are these phenotypes restricted to a small number of cells, even though *Dab1* expression seems more broad. Also, in this context, most of the live migration studies were apparently performed in very rostral neuron populations (Figure 4B) – can the authors clarify the rationale, as the phenotype is most robust at midlevels?

3) In need of text clarifications:

The expression of Reelin in the red nucleus is consistent with a direct role of Reelin signaling in SN-mDA neuronal migration. In the absence of lateral movements, neurons expressing the SN markers and neurons expressing the VTA markers remain intermingled. From the start of the analysis (E13.5), the two populations are well differentiated. Since lateral VTA neurons also express *Dab1* (Figure 1—figure supplement 1), how can the cellular mechanism described in the paper separate two initially intermingled populations of neurons that express the same transducer of Reelin signals? This point deserves to be discussed.

On the issue of morphology: (a) The section heading "mDA adopt a bipolar morphology during moderate/fast phases of migration" is misleading. Numerically, most bipolar cells are not "fast". (Figure 8E,F, show% , but total numbers should be mentioned as well); (b) In all these morphological studies, are these migrating cells extending axons?

The mutant phenotype documented here is rather modest and it should not be overinterpreted. The title claims that that formation of the SN requires Reelin signaling and mDA neuron lateral migration. But SN does appear to form in mutant mice, despite some mingling of lateral and medial mDA neurons.

The authors should discuss whether the deficit of mDA neuron lateral migration in Reelin and Reelin signaling mutant mice disrupts SN function or connectivity. Even if it doesn't, it would be interesting to speculate about the functional significance of this process and thus the importance of Reelin signaling in the formation of this midbrain structure.

"We show that Reelin signaling promotes laterally-biased movements in mDA neurons during their slow migration mode, stabilizes leading process morphology and increases the probability of fast, laterally-directed migration." Related to this point, one could argue that a small population of "fast" neurons, in the *Dab1* CKO, were now shifted into the "slow" bin (as shown in Figure 5), and it is these neurons that are aberrant and result in the appearance of an effect on laterally biased movements in slow migrating neurons (Figure 7).

---

## [Author Response]

Summary:In this study the authors examined SN formation in the midbrain, and specifically examined the lateral migration of mDA neurons away from the midline and the VTA. Using mutant mice, the same investigators and others previously showed that Reelin signaling is required for this particular event. Here using new Dab1 conditional knock out mice in DA neurons they demonstrate that laterally migrating mDAs cell-autonomously require Reelin signaling; they then characterize in detail the modes of migration of these neurons and establish that Dab1 is required to promote movement and especially fast, laterally-oriented migration. Finally, they show that bipolar morphology correlates with this fast mode of migration.The paper clearly identifies and characterizes a cellular behavior responsible for the spatial segregation of two populations of DA neurons and the formation of two nuclei with distinct functions. Overall, the study is solid, employing advanced genetic and imaging techniques to carefully document the lateral migration of mDA neurons, and the specific requirement of Dab1 (and by implication Reelin signaling) in this process. The rigor of this descriptive study will lay a strong foundation for future studies that address mechanism in greater detail.

We thank the reviewers for their insightful comments. We have improved the figures and manuscript accordingly. We analyzed the SN and VTA in *Dab1* CKO and *Dab1^-/-^* mice at several rostrocaudal levels, and also quantified the net loss in SN and the corresponding net gain in the VTA at three rostrocaudal levels. We provide new data on the projections of control, *Dab1* CKO and *Dab1^-/-^* mDA neurons at developmental and adult time-points. To assess microtubule organization and stability in presence and absence of Reelin signaling, we performed immunostaining for α-tubulin, acetylated α-tubulin and EB3 in midbrain sections of control and *Dab1^-/-^* embryos.

Essential revisions:The reviewers appreciated the careful descriptive nature of the study and agreed that substantially advancing the mechanism is likely beyond the scope of the current study. Nonetheless, there were two experiments that the reviewers felt would strengthen the study within its current bounds, and several points that could be addressed in the text. These are as follows: 1) About cellular mechanism: Wild-type and Dab1-/- SN-mDA neurons strongly differ by their morphology. Immunoblotting did not reveal any differences, but immunoblotting might miss subtle defects at the origin of those migratory abnormalities. In contrast, the organization of the microtubule cytoskeleton, especially the relative abundance of stable and dynamic microtubules could provide clues for understanding the migratory defects observed in Dab1-/- SN-mDA neurons. Such analysis can be performed by immunostaining and will give a precise information.

We analyzed the organization of microtubule cytoskeleton by staining ventral midbrain sections for α-tubulin, acetylated α-tubulin and end binding protein 3 (EB3) (Meseke et al., 2012) in control and *Dab1^-/-^* embryos at E13.5 (same stage as the organotypic slices). Co-staining with TH revealed slight variability in the intensity of α-tubulin, acetylated α-tubulin and EB3 within SN-mDA neurons within both groups (n=3 embryos for control and *Dab1^-/-^*). Based on these three markers, we could not identify any obvious differences in the microtubule organization of SN-mDA neurons in control and *Dab1^-/-^* embryos (Figure 9—figure supplement 2, Author response images 1-3). Author response images 1-3 include images of the cortex to demonstrate that the immunostaining for α-tubulin, acetylated α-tubulin and EB3 was specific. These data (Figure 9—figure supplement 2) are described in the Results (subection “Reelin downstream signaling in the ventral midbrain”). The caveats of the current analysis are mentioned in the Discussion (subsection “Reelin downstream signaling in SN-mDA neurons”).

**Author response image 1. respfig1:** α-tubulin in control and *Dab1^-/-^* E13.5 brains. (A-I) TH (A,D,G) α-tubulin (B,E,H) and overlap (C,F,I) in the developing mDA domain of control embryos (n = 3, from 2 litters). (C,F,I) Overlap of TH and α-tubulin in controls. (JR) TH (J,M,P), tubulin (K,N,Q) and overlap panels (L,O,R) in in *Dab1^-/-^* midbrains. Selected region is comparable to the one shown in controls (n = 3, across 2 litters). Control Embryo #1, *Dab1^-/-^* Embryo #3 is also shown in Figure 9—figure supplement 2. No obvious phenotype could be identified based on this analysis. (S,T) α-tubulin in the cortex for comparison.

**Author response image 2. respfig2:** Acetylated α-tubulin in control and *Dab1^-/-^* E13.5 brains. (A-I) TH (A,D,G) and acetylated α-tubulin (B,E,H) in the developing mDA domain of control embryos (n = 3, from 2 litters). (C,F,I) overlap of TH and acetylated α-tubulin in controls. (J-R) TH (J,M,P), acetylated α-tubulin (K,N,Q) and overlap panels (L,O,R) in *Dab1^-/-^* midbrains. Selected region is comparable to the one shown in controls (n = 3, across 2 litters). Control Embryo #1, *Dab1^-/-^* Embryo #3 is also shown in Figure 9—figure supplement 2. No obvious phenotype could be identified based on this analysis. (S,T) Acetylated α-tubulin in the cortex for comparison.

**Author response image 3. respfig3:** EB3 in control and *Dab1^-/-^* E13.5 brains. (A-I) TH (A,D,G) EB3 (B,E,H) and overlap (C,F,I) in the developing mDA domain of control embryos (n = 3, from 2 litters). (C,F,I) Overlap of TH and EB3 in controls. (J-R) TH (J,M,P), EB3 (K,N,Q) and overlap panels (L,O,R) in *Dab1^-/-^* midbrains. Selected region is comparable to the one shown in controls (n = 3, across 2 litters). Control Embryo #1, *Dab1^-/-^* Embryo #3 is also shown in Figure 9—figure supplement 2. No obvious phenotype could be identified based on this analysis. (S,T) EB3 in the cortex for comparison.

2) About the details of the DA populations involved in the phenotype: The authors mention that the phenotype is most severe at mid levels. In general, even at mid levels, the phenotype is subtle with only approximately 10-15% of the neurons being mislocalized (Figure 3C, bin 1 and 3). Is there even a phenotype in more rostral sections? The Dab1 CKO should be analyzed/quantified along the rostrocaudal axis. If one counts total SN and VTA neurons (rather than mediolateral bins), what is the net reduction in SN, and what is the net gain in VTA DA neurons?

We now provide a quantification of control and the Dab1 CKO animals at three rostrocaudal levels of the midbrain containing both the SN and VTA at P21-30 (Figure 1—figure supplement 4). Reduction in the number of mDA neurons in the SN and the corresponding gain in the region of the VTA has been quantified for three rostrocaudal levels individually and in total (Figure 1—figure supplement 4). The results show a significant loss of mDA neurons in the SN and a corresponding gain in the VTA at the intermediate level and when the three rostrocaudal levels are combined. We also added images of the SN at rostral levels (Figure 1—figure supplement 4A-C) to show the disorganization of the SN at this level. A description of the results was added to the text (Results subsection “Reelin signaling acts directly on tangentially migrating mDA neurons”).

A related question is why are these phenotypes restricted to a small number of cells, even though Dab1 expression seems more broad. Also, in this context, most of the live migration studies were apparently performed in very rostral neuron populations (Figure 4B) – can the authors clarify the rationale, as the phenotype is most robust at midlevels?

*Dab1* expression: Please see point 3 for an explanation on how *Dab1* expression is consistent with the phenotype observed in the absence of *Dab1*.

Live migration studies: As horizontal slices were used for the imaging, our analysis was not restricted to a particular rostrocaudal level. We added a sentence to clarify this in the Results (subsection “Time-lapse imaging of tangentially migrating mDA neurons reveals diverse migratory behaviors across a population of neurons, and in individual neurons across time”).

3) In need of text clarifications:The expression of Reelin in the red nucleus is consistent with a direct role of Reelin signaling in SN-mDA neuronal migration. In the absence of lateral movements, neurons expressing the SN markers and neurons expressing the VTA markers remain intermingled. From the start of the analysis (E13.5), the two populations are well differentiated. Since lateral VTA neurons also express Dab1 (Figure 1—figure supplement 1), how can the cellular mechanism described in the paper separate two initially intermingled populations of neurons that express the same transducer of Reelin signals? This point deserves to be discussed.

*Dab1* expression: We have previously shown that *Dab1* mRNA is localized to the most lateral cell population in the developing mDA domain at E13.5 (Bodea et al., 2014, Figure 6), consistent with a specific response of laterally migrating cells to Reelin signaling. In Figure 1—figure supplement 2, we now present immunostaining for DAB1 at E13.5 confirming that DAB1 is expressed in laterally but not medially located mDA neurons at this stage. At E15.5, *Dab1* is expressed more broadly, both at the mRNA (Bodea et al., 2014, Figure S5) and protein level (Figure 1—figure supplement 2). However, at this time point SN and VTA populations are essentially separated. Whether *Dab1* has additional functions in mDA development at E15.5 and later warrants further studies. The change in the *Dab1* expression pattern between E13.5 and E15.5 is now described in the Results (subsection “Reelin signaling acts directly on tangentially migrating mDA neurons”).

Intermingling of mDA neurons: We would like to point out that we do not try to claim that Reelin is the key factor for the sorting out of SN from VTA cells. Based on our data, Reelin rather seems to regulate the subsequent lateral migration step. The intermingling of SN and VTA mDA neurons in *Dab1* CKO described in Figure 3 only occurs at the SN/lateral VTA border and is likely a consequence of the inability of *Dab1^-/-^* cells to undergo this lateral migration step. We clarify this in the Results (subsection “Reelin signaling contributes to the segregation of SN- and VTA-mDA neurons into separate clusters”) and the Discussion (subsection “Reelin signaling directly regulates tangential migration of SN-mDA neurons”).

On the issue of morphology: (a) The section heading "mDA adopt a bipolar morphology during moderate/fast phases of migration" is misleading. Numerically, most bipolar cells are not "fast". (Figure 8E,F, show% , but total numbers should be mentioned as well); (b) In all these morphological studies, are these migrating cells extending axons?

a) We changed the title of this section to better reflect the data (subsection “mDA neurons are predominantly associated with bipolar morphology during moderate-to-fast phases of migration”). Figure 8E,F refers to slow, moderate and fast phases of movement. We have corrected the Y-axis in these histograms to clarify this. Note that Figure 8E,F are now in Figure 8—figure supplement 2 and that histograms are shown separately for fast, moderate and slow migration phases. We added the total number of data points to the legend of Figure 8—figure supplement 2.

b) Projections of mDA neurons dive ventrally into the developing hypothalamus in both control and *Dab1^-/-^* brains as now illustrated with TH whole-mount staining and TH immunostaining on coronal sections (Figure 3—figure supplement 1A-D, Results subsection “Reelin signaling contributes to the segregation of SN- and VTA-mDA neurons into separate clusters”). Thus, it is difficult to reliably visualize axonal projections in horizontal slices used for live-imaging (see also Bodea and Blaess, 2012).

The mutant phenotype documented here is rather modest and it should not be overinterpreted. The title claims that that formation of the SN requires Reelin signaling and mDA neuron lateral migration. But SN does appear to form in mutant mice, despite some mingling of lateral and medial mDA neurons.

We have changed the title of the paper to account for this point (“Correct setup of the substantia nigra…”)

The authors should discuss whether the deficit of mDA neuron lateral migration in Reelin and Reelin signaling mutant mice disrupts SN function or connectivity. Even if it doesn't, it would be interesting to speculate about the functional significance of this process and thus the importance of Reelin signaling in the formation of this midbrain structure.

We analyzed the density of TH^+^ projections to the striatum in controls and *Dab1* CKO animals at P30 and found no significant difference between the two groups (Figure 3—figure supplement 1E-G, Results subsection “Reelin signaling contributes to the segregation of SN- and VTA-mDA neurons into separate clusters”). A more detailed insight into possible alterations in VTA and SN projections would require further experiments (e.g. specific tracing and/or optogenetic activation of SN neurons, single cell labeling to visualize axonal arbors). This has been added to the Discussion (subsection” Reelin signaling directly regulates tangential migration of SN-mDA neurons”).

"We show that Reelin signaling promotes laterally-biased movements in mDA neurons during their slow migration mode, stabilizes leading process morphology and increases the probability of fast, laterally-directed migration." Related to this point, one could argue that a small population of "fast" neurons, in the Dab1 CKO, were now shifted into the "slow" bin (as shown in Figure 5), and it is these neurons that are aberrant and result in the appearance of an effect on laterally biased movements in slow migrating neurons (Figure 7).

Based on our data we cannot resolve this point. We have added a sentence to the Discussion to reflect this (subsection “Reelin promotes a preference for directed migration”).